# Exploring penetrance of clinically relevant variants in over 800,000 humans from the Genome Aggregation Database

Sanna Gudmundsson [1,2,3,4] ✉, Moriel Singer-Berk[1], Sarah L. Stenton [1,2,3], Julia K. Goodrich[1], Michael W. Wilson[1], Jonah Einson[5], Nicholas A. Watts[1], Genome Aggregation Database Consortium*, Tuuli Lappalainen [4,5], Heidi L. Rehm[1,2], Daniel G. MacArthur[1,6,7] & Anne O'Donnell-Luria [1,2,3] ✉

Incomplete penetrance, or absence of disease phenotype in an individual with a disease-associated variant, is a major challenge in variant interpretation. Studying individuals with apparent incomplete penetrance can shed light on underlying drivers of altered phenotype penetrance. Here, we investigate clinically relevant variants from ClinVar in 807,162 individuals from the Genome Aggregation Database (gnomAD), demonstrating improved representation in gnomAD version 4. We then conduct a comprehensive case-by-case assessment of 734 predicted loss of function variants in 77 genes associated with severe, early-onset, highly penetrant haploinsufficient disease. Here, we identify explanations for the presumed lack of disease manifestation in 701 of 734 variants (95%). Individuals with unexplained lack of disease manifestation in this set of disorders are rare, underscoring the need and power of deep case-by-case assessment presented here to minimize false assignments of disease risk, particularly in unaffected individuals with higher rates of secondary properties that result in rescue.

Accurately predicting disease risk in asymptomatic individuals, for example in prenatal diagnosis as well as cancer and other later-onset disorders, is critical for realizing precision medicine. Incomplete penetrance, defined as the absence of the disease phenotype in individuals with a disease-associated variant, presents an added challenge in predicting risk and determining variant pathogenicity, especially in yet unaffected individuals. Genome sequencing of apparently healthy individuals frequently identifies presumed disease-causing variants in the absence of reported disease[1–5]. This could suggest that incomplete penetrance is either a surprisingly common feature of human disease or alternatively may be driven by limitations in the accuracy of variant-calling and annotations.

Disease penetrance and its underlying mechanisms are not well understood. Historically, penetrance estimates have relied on symptomatic proband-identified disease cohorts, resulting in overestimating disease risk due to ascertainment bias[6]. Analysis of large-scale population databases and biobanks can provide a less biased avenue to investigate penetrance[7]. Recent penetrance studies utilizing population data have focused on assessing variant penetrance for specific disorders, e.g., prion disease[8], metabolic conditions[9], and maturity-onset diabetes of the young (MODY)[10], and more broadly

[1]Program in Medical and Population Genetics, Broad Institute of MIT and Harvard, Cambridge, MA, USA. [2]Center for Genomic Medicine & Analytic and Translational Genetics Unit, Massachusetts General Hospital, Boston, MA, USA. [3]Division of Genetics and Genomics, Boston Children's Hospital, Harvard Medical School, Boston, MA, USA. [4]Science for Life Laboratory, Department of Gene Technology, KTH Royal Institute of Technology, Stockholm, Sweden. [5]New York Genome Center, New York, NY, USA. [6]Centre for Population Genomics, Garvan Institute of Medical Research and UNSW Sydney, Sydney, New South Wales, Australia. [7]Centre for Population Genomics, Murdoch Children's Research Institute, Melbourne, Australia.*A list of authors and their affiliations appears at the end of the paper. ✉e-mail: sgudmund@broadinstitute.org; odonnell@broadinstitute.org

investigating the association of pathogenic variants with 401 phenotypes in 379,768 individuals from the UK Biobank[5]. Investigations moving beyond descriptive reports and association studies towards focusing on mechanisms underlying incomplete penetrance are few and disease-specific[11–14], with examples of how expression levels[15], eQTL[16], and sQTLs[17] could modulate penetrance. However, a statistical approach to investigate eQTL association with incomplete penetrance in neurodevelopmental disease in 1,700 trios from the Deciphering Developmental Disorder cohort did not find altered gene expression due to known eQTLs to be an explanation for incomplete penetrance in the unaffected parents carrying the variant of the affected probands[18].

The Genome Aggregation Database (gnomAD) is a widely used publicly available collection of population variant data, currently sharing harmonized data from 807,162 individuals, including 76,215 genomes and 730,947 exomes (version 4, released November 2023). The gnomAD dataset has played a key role in supporting the discovery of genes and variants associated with genetic diseases, enabling improved variant classification in clinical as well as mechanistic-focused interpretation of variants in disease-discovery research settings[19–21]. Investigation of well-established and predicted disease-associated variants in unaffected individuals in gnomAD presents an unexplored opportunity to improve our variant interpretation abilities, increase understanding of variant effect, and inform on mechanisms affecting the penetrance of disease. The unprecedented size, rigorous quality control pipelines with joint variant-calling over all samples, and diverse ancestry make gnomAD an excellent resource for large-scale analysis of variants reported as pathogenic in clinical databases (e.g., ClinVar). Because phenotype data for individuals in gnomAD is not systematically collected nor able to be shared, we focus on variants associated with severe, dominantly inherited diseases not typically found in individuals recruited for common disease studies or biobanks, from which gnomAD samples originate.

ClinVar is a publicly accessible database that collects submissions mainly from diagnostic laboratories and some research studies, including the clinical significance of variants, and optionally, the associated disease, as well as applied evidence[22]. The open submission model allows collection of a massive variant classification dataset, which powers large-scale population-based studies; to date, over 3.6 million records have been submitted, and over 2.4 million unique variants have been assigned a clinical significance of pathogenic (P), likely pathogenic (LP), uncertain significance (VUS), likely benign (LB), or benign (B)[22], with most submitters using recommended standards from the American College of Medical Genetics and Genomics (ACMG) and the Association for Molecular Pathology (AMP)[23]. An inevitable consequence of a crowd-sourced, voluntary, point-in-time submission model is the risk of outdated and/or inaccurate pathogenicity classifications, hence these data must be interpreted cautiously. Having multiple submitters agree on classification and sharing the evidence used towards the classification builds confidence, but 77.3% of variants are currently classified by only one laboratory[24]. Although the vast majority of P/LP variants in any population database are likely to be variants causing disease in an autosomal recessive (AR) manner and any carrier of one allele would simply be a carrier of AR disease, there are also observations of P/LP variants reported to cause disease in an autosomal dominant (AD) manner. Naturally, some of these variants will be expected in common disease studies or biobanks due to being hypomorphic or associated with mild phenotypes, having known incomplete penetrance or variable expressivity, or being late-onset conditions that have not manifested at the time of enrollment, but some are expected to be much less common or even absent in population databases due to the nature of the phenotype (severe, early-onset, highly penetrant).

Loss of function (LoF) variants (here including nonsense, essential splice site, and frameshift variants) have important implications in disease biology and are an especially interesting group of variants from an incomplete penetrance perspective as they are considered to have a fairly uniform variant-to-function effect through nonsense-mediated mRNA decay. Thus, any true LoF variant in a haploinsufficient disease-associated gene is expected to result in disease. In recent work, we provided a protocol for improved evaluation and classification of predicted LoF (pLoF) variants and demonstrated that deeper pLoF assessment beyond standard annotation pipelines is crucial to reduce pLoF false-positive classification rates[21]. The framework presented a set of 32 rules designed based on previously accepted mechanisms where a variant annotated as pLoF by VEP does not result in loss of the protein product, here referred to as pLoF evasion mechanisms. This includes identifying local modifying variants, assessing the biological relevance of the site, but also evaluating for evidence of a variant being an artifact (a challenge in population datasets where variants have not been analytically validated by an orthogonal method). Each pLoF variant is labeled according to the criteria of these rules, which adds up to a final verdict of LoF, Likely LoF, Uncertain LoF, Likely not LoF, or Not LoF for each pLoF variant[21]. Further study of pLoF variants associated with disease in presumably unaffected individuals in population databases can enhance our understanding of pLoF functional impact and highlight the possibility of incomplete penetrance in the investigated conditions.

In this study, we explore disease-associated variants in 807,162 individuals from gnomAD to increase understanding of the underlying reason for tolerance of presumed disease-causing variants and mechanisms of incomplete penetrance. Specifically, we have (1) explored the landscape of ClinVar variants in these individuals, including investigation of how representation varies between gnomAD releases, (2) investigated the prevalence of modifying variants as an explanation for incomplete penetrance in a subset of P/LP variants, (3) deeply investigated all pLoF variants in 76,215 gnomAD genomes associated with a set of 77 severe, early-onset, highly penetrant haploinsufficiency disorders for lack of disease manifestation due to limitation in variant annotation, calling somatic variants, detecting artifacts and by mechanisms of incomplete penetrance in truly pathogenic variants. These large-scale analyses of presumed disease-causing variants in the general population provide valuable insight into disease-variant interpretation and the identification of modifying variants that can result in incomplete penetrance.

## Results

### Improved representation of ClinVar variants in gnomAD v4

We investigated the extent to which variants reported in ClinVar are also represented in gnomAD. Of 2,314,231 unique ClinVar variants, filtered to include all single nucleotide variants and indels (<50 base pairs) with assigned clinical significance (P/LP, VUS, B/LB or conflicting classifications), 1,702,421 variants (73.6%) were present in at least one of 807,162 individuals in gnomAD. As expected, we observed a lower representation of P/LP ClinVar variants 66,571/221,975 (30.0%) and a higher representation of VUS, B/LB, and variants with conflicting classifications (73.1%, 83.6%, and 88.8%; Fig. 1a–b). P/LP variants found in gnomAD are of more deleterious variant classes (e.g., pLoF variants: nonsense, frameshift, and essential splice variants), compared to B/LB that are of less deleterious variant classes (e.g., intronic and synonymous variants). VUS mostly consists of missense variants (Fig. 1c). In addition to a lower representation of ClinVar P/LP variants in gnomAD, P/LP variants are rarer compared to other classes. 97.6% of P/LP variants have an allele frequency (AF) of less than 0.01% (AF < 0.0001, Fig. 1d, and Supplementary Data S1); 61.1% are observed in five or fewer individuals, and 29.8% in a single individual in gnomAD (Fig. 1e, and Supplementary Data S2). The 66,571 P/LP variants are found on 8,110,001 alleles in 807,162 individuals, resulting in an average of 10.0 pathogenic alleles per individual. Of 66,571 P/LP variants, 63,646 variants could be assigned an inheritance pattern based on the reported

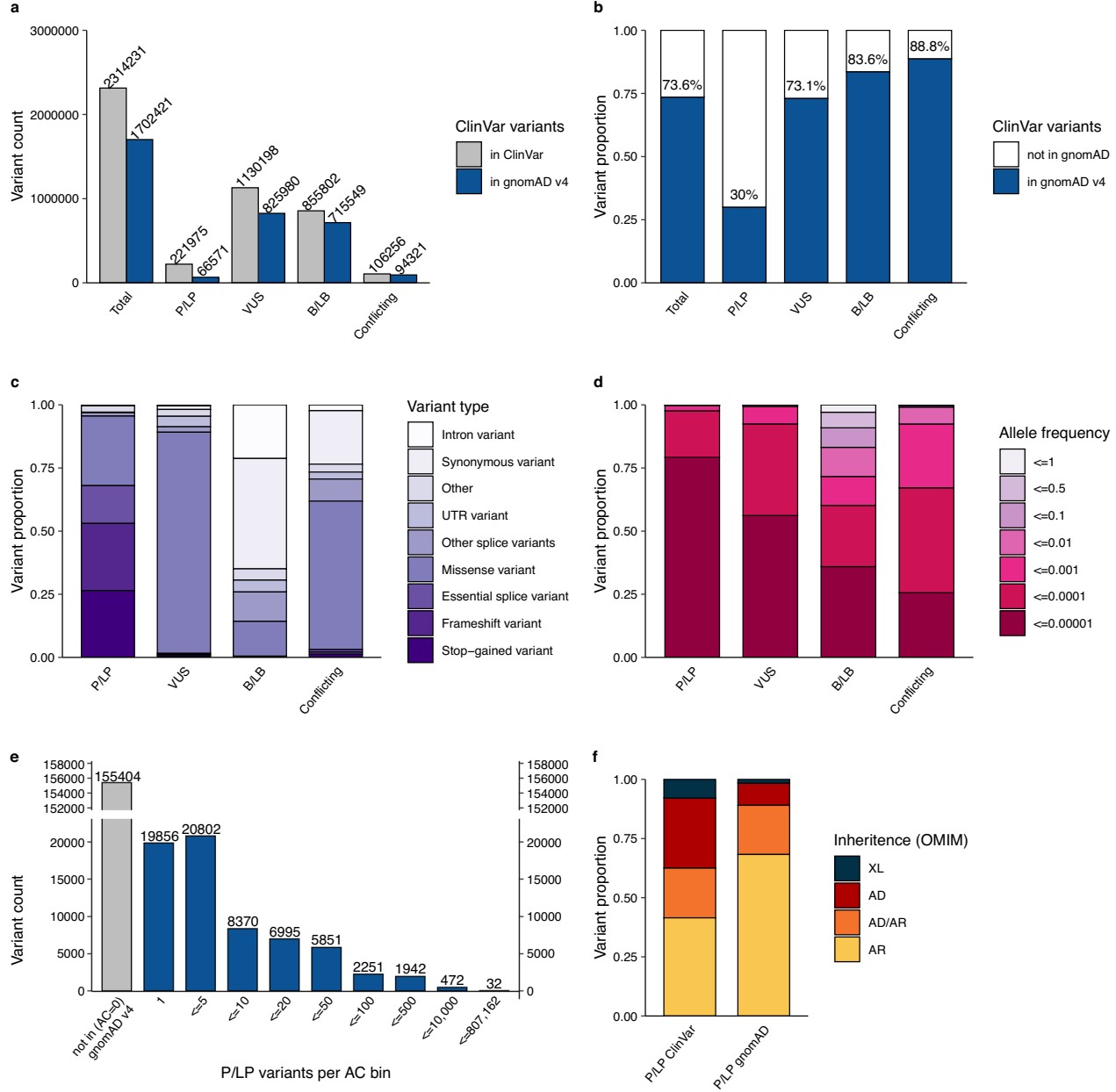

**Fig. 1 | Representation of ClinVar variants in 807,162 individuals in gnomAD v4.**
**a** Variant count of ClinVar variants in ClinVar (gray) vs. gnomAD (blue) in each classification category pathogenic/likely pathogenic (P/LP), variant of uncertain significance (VUS), benign/likely benign (B/LB) or with conflicting classifications. **b** Percentage of ClinVar variants reported in gnomAD in at least one individual. **c** The proportion of ClinVar variants in gnomAD by variant type within each clinical significance classification and, **d** by allele frequency bin. **e** Total number of P/LP variants within each allele count bin (including variants absent from gnomAD for comparison). **f** The inheritance pattern of the gene harboring the P/LP variants in gnomAD versus all variants in ClinVar, X-linked (XL), Autosomal dominant (AD), Autosomal recessive (AR).

inheritance pattern in OMIM. As expected, P/LP variants found in gnomAD are primarily found in genes associated with disorders with an AR inheritance pattern, and there is an underrepresentation of P/LP variants in genes associated with disorders with an AD inheritance pattern compared to variants reported in ClinVar (Fig. 1f).

The 5.7-fold increase from 141,456 to 807,162 individuals from gnomAD v2 to v4 has resulted in improved representation of unique ClinVar variants, from 56.9% in v2 to 73.6% in v4. Representation of P/LP variants has close to doubled and increased from 16.3% to 30.0%, and B/LB variants have increased from 70% to 83.6% (Fig. 1a–b, and Figure S1a–b). Reassuringly, distributions of variant types, AF, allele counts (AC), and inheritance are largely consistent between the

datasets (Fig. 1c–f, and Figure S1c–f), with a trend towards ClinVar variants having a lower AF as gnomAD population size increases, i.e., an observed lower AF in v4 compared to v2 (Figure S2). The average number of P/LP variants per individual is similar between versions, 8.8 per individual in gnomAD v2 (1,240,951 alleles in 141,456 individuals) compared to 10.0 per individual in gnomAD v4.

## Example of genetic ancestry group-specific incomplete penetrance

We investigated if a set of 3957 P/LP variants found in 31,014 individuals were tolerated due to modifying pLoF variants in the same gene, potentially reducing the expression of the pathogenic allele. We

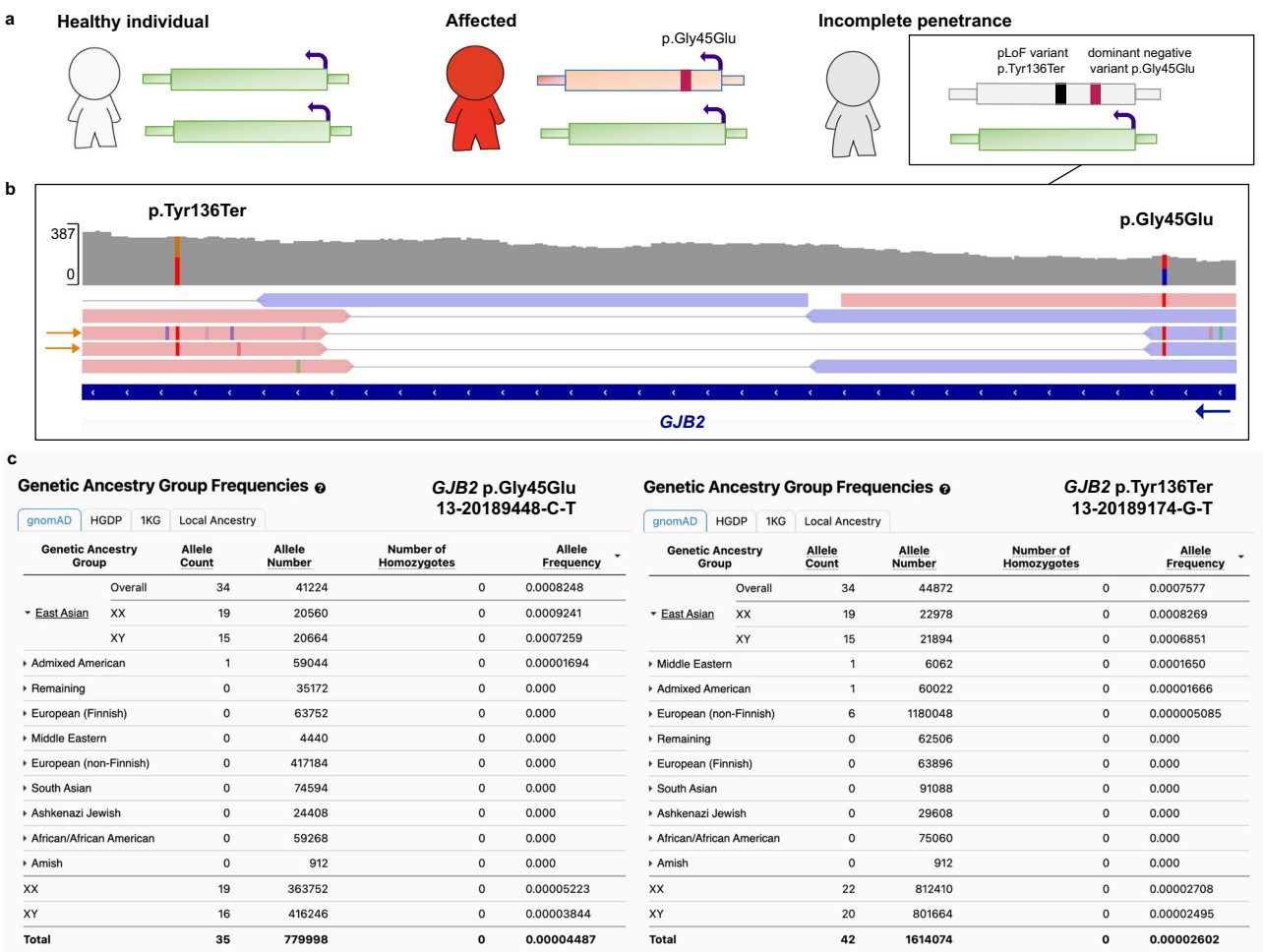

**Fig. 2 | The p.Gly45Glu *GJB2* variant associated with severe pediatric disease is rescued by a modifying variant in the East Asian genetic ancestry group.**
**a** Schematic of the rescue mechanisms where the modifying p.Tyr136Ter nonsense variant *in cis* with the dominant-negative disease-causing p.Gly45Glu missense variant, results in incomplete penetrance by mono-allelic expression of the reference allele only (green). **b** Paired individual-level read data of the two variants occurring *in cis*. **c** East Asian (*n* = 34) and Admixed American (*n* = 1) individuals carrying both the pathogenic p.Gly45Glu variant (left panel) and the p.Tyr136Ter modifying variant (right panel).

included all P/LP ClinVar variants in gnomAD with AC ≤ 50, located in a gene not constrained for LoF (predicted loss-of-function intolerance < 0.9, pLI), and with at least one condition of AD inheritance in OMIM, suggesting that pathogenicity is more likely to act through a gain-of-function mechanism.

We found one example of pLoF variant modifying disease penetrance of a pathogenic variant in *GJB2* p.Gly45Glu (13-20189448-C-T) that is reported to cause a severe form of keratitis-ichthyosis-deafness syndrome but found in 35 individuals in gnomAD v4. The condition is lethal due to severe skin lesions, infections, and septicemia[25,26], through a dominant negative effect resulting in disturbed ion channel transportation. Our analysis confirmed that all 35 individuals also had a downstream nonsense p.Tyr136Ter variant (13-20189174-G-T). Access to paired individual-level sequencing read data from one individual confirmed that the two variants were present *in cis*. This specific p.Tyr136Ter[27], as well as many other nonsense variants, is associated with AR hearing loss suggesting that the impact of the combined p.Gly45Glu and p.Tyr136Ter allele in these individuals is converted to loss-of-function. The haplotype is likely identical by descent with 34 out of 35 gnomAD individuals belonging to the East Asian genetic ancestry group (Fig. 2). Our analysis suggests that this is a rare example of incomplete penetrance as we did not find evidence of this being a common mechanism in the general population (report of results in Supplementary Note, Supplementary Data S3).

## High rate of rescue identified for presumed disease-causing pLoF variants in genes associated with dominant disease

We sought to assess pLoF variants in genes associated with disease under the hypothesis that modifying variants could account for some observations of incomplete penetrance (Fig. 3a–b). We selected 77 haploinsufficient genes associated with severe, highly penetrant, early onset disorders (before the age of three years), expected to be absent or rarely seen in common disease studies or biobanks (e.g., gnomAD). We used de novo rate as a proxy for penetrance (Supplementary Data S4). In these 77 genes, investigating 807,162 individuals, we found 4,464 pLoF variants, of which 3,957 were high-confidence pLoF variants by Loss-Of-Function Transcript Effect Estimator (LOFTEE)[4]. Of 3,957 high-confidence pLoF, 3,223 are from exomes (81%) and 734 in genomes (19%). gnomAD v4 includes 9.4% (76,215) genomes and 90.6% (730,947) exomes, hence we observed a higher rate of these unexpected pLoF in genomes (19%) compared with exomes (81%) (Figure S3). The majority (87%) of these variants are extremely rare with an AC of five or less (Figure S4).

We performed deep case-by-case assessments on the 734 high-confidence pLoF variants found in gnomAD genomes, building on a framework for pLoF variant assessment[21] using conservative rules (Supplementary Data S5) to exclude variants likely to not result in LoF (Supplementary Data S6). Each pLoF variant was assigned a verdict of "Not LoF/Likely not LoF" when we found evidence the

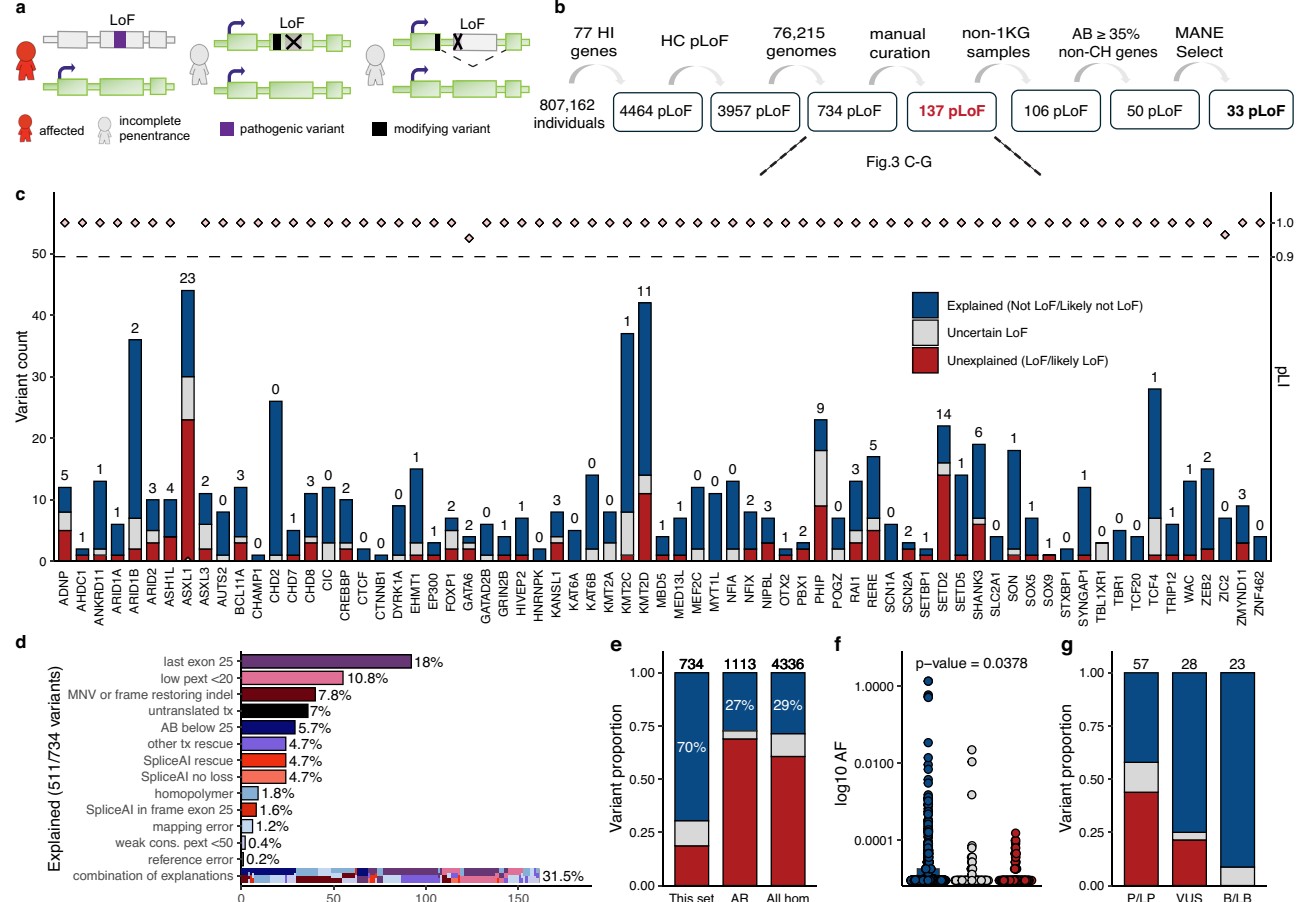

**Fig. 3 | Deep assessment of 734 predicted high-confidence (HC) loss-of-function (pLoF) variants found in 77 haploinsufficient (HI) genes in 76,215 genomes.** **a** Schematic of how modifying variants (black) may result in lack of a phenotype. **b** Filtering approach. 1KG: 1000 Genomes Project, AB: Allele balance, CH: clonal hematopoiesis, MANE: Matched Annotation from NCBI and EMBL-EBI. **c** Variant count per HI gene colored by (also for e-g): explained (blue), uncertain (gray), unexplained (red). The number indicates unexplained variants (red). **d** The explanation for evading LoF in 511 of 734 (69.6%) of variants, MNV: multi-nucleotide variant, pext: per-base-expression score, tx: transcript, cons: conservation, 'last

exon 25' and 'SpliceAI in frame exon 25': less than 25% protein removed; 'AB below 25': allele balance <25%. **e** Comparison of outcome in different gene sets, this set (left), heterozygous pLoF variants in autosomal recessive (AR) disease genes in gnomAD v2 (middle), all homozygous (hom) pLoF variants in gnomAD v2 (right). **f** Allele frequency (AF) of variants in this set by LoF curation outcome ($\log_{10}$ scale), lower AF for unexplained ($n = 137$) than explained (n = 511), $p = 0.0378$ two-sided Student's t-test. **g** The proportion of pLoF variants explained, uncertain, or unexplained within each ClinVar clinical classification category pathogenic/likely pathogenic (P/LP), uncertain significance (VUS), benign/likely benign (B/LB).

variant would not result in ablated expression of that protein (evade LoF), "Uncertain LoF" when there was partial or conflicting evidence or lack of data to assess the variant, and "LoF/Likely LoF" when we did not identify a reason for evading pLoF using this protocol. We defined 511 of 734 (69.6%) pLoF variants as explained (Not LoF/Likely not LoF; Fig. 3, blue). The most common explanations were: location in the last exon or last 50 base pairs of the penultimate exon, resulting in a variant not predicted to undergo nonsense-mediated decay (may or may not be pathogenic but excluded from this analysis; 18%); location in a region with low per-base expression score (pext, an average expression score derived from GTEx adult post-mortem tissues; 10.8%); rescue by modifying variants (multi-nucleotide variants or frame-restoring indels; 7.8%), location in a transcript isoform containing a stop-codon (unlikely to be coding; 7%); allele balance (AB) for the alternate allele below 25% (indicating the variant may be somatic; 5.7%), other transcript rescues (including transcripts with downstream methionine within the first exon, unconserved alternate open reading frame, and variants in overhang exons 4.7%[21]), and rescue by different types of splice modifying variants predicted by SpliceAI (rescue by skipping/deletion of an in-frame exon, in-frame up- or down-stream alternative splice site 4.7%, and no loss detected 4.7%). A large portion of variants (31.5%) were explained by

multiple ("combination of explanations") of these rescue modes (Fig. 3d, and Figure S5). We labeled 86 of 734 (11.7%) pLoF variants as uncertain LoF (Fig. 3, gray), due to lack of read data limiting the ability to analyze for local modifying variants, concern for sequencing errors, contradictory SpliceAI predictions, or other types of weaker evidence of not being LoF (Figure S6, Supplementary Data S5). For 137 of 734 (18.7%) pLoF variants, the reason for lack of disease manifestation in 236 individuals was still unexplained and they remained interesting candidates for further investigation (Fig. 3, red, and Supplementary Data S6).

The observed pLoF evasion rate of 69.8% for this set of pLoF variants is notably higher than other sets of pLoF variants assessed in previous work reporting 27.3% (304/1113) evasion for heterozygous pLoF variants associated with 22 AR disorders[21], and 28.7% (1245/4336) in a set including all homozygous pLoF variants in gnomAD v2[4] (Fig. 3e). The AF was higher for variants where LoF evasion could be explained compared to variants where the reason was yet to be found ($p = 0.0378$, two-sided Student's t-test; Fig. 3f).

We assessed how well the explained (blue) versus unexplained (red) variants align with reported ClinVar pathogenicity classifications. ClinVar classifications (B, LB, VUS, LP or P) were available for 108 of 734 variants; B/LB variants were more likely to be explained compared

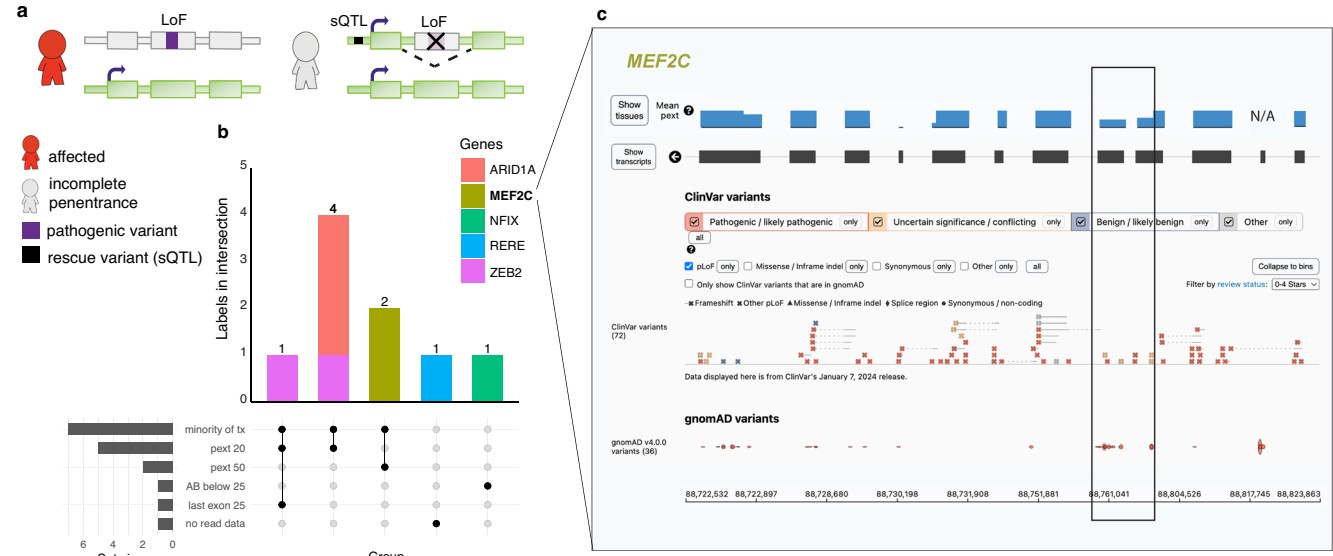

**Fig. 4 | Splicing quantitative trait loci (sQTL) analysis. a** Schematic of alternative splicing as a mechanism for incomplete penetrance. **b** Labels for each of the nine predicted loss of function (pLoF) variants. Bars are colored by gene, labels indicate variant count per gene. Tx: transcript, pext: per base expression score, AB: allele balance. **c** Overview of pLoF variants in *MEF2C* in ClinVar and in gnomAD v4 (all variants, including exomes). Black box marks region of reduced mean pext score (from v2, blue bar, for one exon the score is not available [N/A]) and enrichment of *MEF2C* pLoF variants in gnomAD.

to P/LP variants. The vast majority (21 of 23) of B/LB variants' reason for evasion was explained by the protocol, and the remaining 2 of 23 were uncertain and excluded from further analysis. Of the 57/108 ClinVar pLoF variants that were P/LP, we could explain the lack of phenotype in gnomAD individuals for 25/57 variants (43.9%) (Fig. 3g, Supplementary Data S7). Of all variants explained by being located in the last exon, located in low pext regions, or part of an MNVs in the gnomAD individuals, most were B/LB, with less than 25% of these variants being P/LP in ClinVar. In contrast, for variants explained in our individuals due to potentially being sequencing errors (variant in homopolymer region) or of somatic origin (AB of the alternate allele below 25%), most are reported as P/LP (Figure S7), consistent with the expectation that these would result in LoF when they are bonafide germline variants.

After assessment using the LoF classification framework[21], 137 variants in 236 individuals remained without an explanation. We noted a project-specific enrichment with 25.5% of variants ($n = 35$) or 17.4% of samples ($n = 41$), originating from the 1000 Genomes Project, although the 1000 Genomes Project constitutes less than 5% of gnomAD genome samples. Individuals from this cohort were thus excluded due to a concern that these may be cell culture-acquired variants (see discussion). For the remaining 106 variants in 195 individuals, we observed a trend of higher age distribution in these 195 individuals compared to gnomAD genomes in general (Figure S8a-b), consistent with somatic origin which can rise to a high AB due to age-related clonal hematopoiesis. Variants noted to have an appreciable AB but still below the AB for typical germline variants, defined as an alternative allele ratio under 35% (with those under 20% already filtered by gnomAD QC practices and below 25% filtered by pLoF curation protocol) showed a significant skew towards elderly individuals (two-sided Wilcoxon rank sum test, $p = 0.0013$, Figure S8c), which was also observed for variants in clonal hematopoiesis-associated genes (36 genes[28]; two-sided Wilcoxon rank sum test, $p = 0.00027$, Figure S8d). After filtering variants found in clonal hematopoiesis genes and/or with low alternative AB due to possible somatic occurrence, only 50 variants in 104 individuals remained. Of these 50, 17 pLoF variants were not present in Matched Annotation from NCBI and EMBL-EBI (MANE) Select transcripts, suggesting that they might be of less biological relevance, leaving 33 of 734 (4.5%) pLoF variants without an explanation (Figure S9).

## Incomplete penetrance due to alternative splicing mediated by non-coding sQTL variants

We investigated if alternative splicing of the region with a disease-causing pLoF, mediated by a specific sQTL, could explain incomplete penetrance in any of the original 734 high-confidence pLoF variants identified in gnomAD (Fig. 4a). Out of the 734, nine pLoF variants (already labeled using the pLoF curation framework) fell in regions predicted to be alternatively spliced by a specific sQTL variant (Fig. 4b, Supplementary Data S8). Notably, seven of the nine variants are found in regions with reduced pext scores, also suggesting alternative splicing in the general population, highlighting the usefulness of pext scores in assessing alternative splicing. Two *MEF2C* variants were marked as uncertain due to a 50% reduction in pext score and found in four genome-sequenced individuals with European ancestry, 5-88761023-C-A ($n = 3$) and 5-88761110-C-A ($n = 1$). Further investigation confirmed heterozygosity for the sQTL variant 5-89714113-C-G (non-coding) where the alternate allele is predicted to result in exon exclusion in three of four individuals (global AF 0.28, European AF 0.36), yet data for variant phasing was not available (Supplementary Data S9). Investigation of *MEF2C* in all 807,162 individuals (including exomes) in gnomAD v4 displayed clustering of fourteen pLoF variants in this region in combination with seven pLoF reported in ClinVar (four VUS and three P/LP), suggesting possible inaccurate P/LP classifications of the three ClinVar variants or incomplete penetrance of LoF variants in this region in some individuals due to alternative splicing (Fig. 4c, black box). Additionally, variants in *ARID1A* ($n = 3$) and *ZEB2* ($n = 2$) fell in a region with a ~80% reduction in pext score.

## Discussion

Presumably unaffected individuals with variants predicted to be associated with severe, highly penetrant, and early-onset disease present an opportunity to improve understanding of variant pathogenicity, interpretation, and penetrance. Previous studies of disease-variants present in population databases have focused on descriptive reports[1], statistical associations between disease-variants and phenotypes[8–10,14,29], estimated penetrance of specific disorders[8,9,30], gene-specific mechanisms[15] or a specific type of modifying variants (e.g., eQTLs/ sQTLs)[16–18]. We aimed to move beyond previous efforts by

using a large-scale but non-statistical approach, under the hypothesis that some of these modes are rare and need in-depth investigation of the relevant region to allow discovery of the underlying reason for the lack of disease manifestation. We investigated individuals on a case-by-case level and identified a wide range of explanations underlying tolerance for these pLoF variants in a large set of distinct disorders.

Using the gnomAD v4 dataset with variant data from 807,162 individuals, we could confirm that large-scale efforts like gnomAD increasingly include rare clinically relevant variants, underlining that continued aggregation of data, especially focusing on diversity and underrepresented groups, will power variant analysis. Increased sample size provides more accurate predictions of variant rarity, as well as improved coverage of less common variation. Allele frequencies are mostly consistent over gnomAD versions and allele counts for P/LP ClinVar variants and pLoF variants in genes associated with severe disease are consistently low. The more than 5-fold increase in individuals between v2 and v4 does result in more unique P/LP variants as well as higher allele counts in some cases, which is expected with the larger database size and needs to be considered in analysis.

The utility of studying individuals with pathogenic variants that do not present with the associated phenotype, and the importance of diverse ancestry representation in any population database, was demonstrated by a case of genetic ancestry group-specific incomplete penetrance in East Asians of a *GJB2* variant associated with a lethal form of KID syndrome. We note that although bi-allelic loss of *GJB2* is associated with AR hearing loss, there is a high observed/expected number of pLoF variants in *GJB2* in gnomAD, with a LOEUF score of 1.97, higher than 99.9% (5/3064) of AD and/or AR genes, suggesting a possible selective advantage of pLoF variants in *GJB2* which has been explored previously[31–33].

We investigated all pLoF found in 76,215 v4 genomes in 77 genes associated with autosomal dominant, severe, highly penetrant, early-onset disease. As expected, 76 of 77 genes were loss-of-function constrained with a pLI ≥0.9 in gnomAD v4. *ASXL1* has a pLI of 0.0 in both v4 and v2 due to somatic variants rising to higher AB due to clonal hematopoiesis and deflating the pLI constraint score[2]. Of 734 pLoF variants identified in the 77 genes, we found an explanation for the lack of disease manifestation and tolerance of the pLoF variant for 95% of variants. The explanations identified highlighted a combination of our current limitations in pLoF annotations, the occurrence of somatic variation, rare instances of sequencing artifacts, and examples of incomplete penetrance of pathogenic variants. The most common explanations were location in the last exon (or last 50 base pairs of the penultimate exon), low pext region, rescue by nearby secondary variants (MNVs/frame restoring indels or splice variants), and somatic variants resembling germline variants due to clonal expansion resulting in elevated AB of the alternate allele. In general, a pLoF variant in this set of genes associated with highly penetrant conditions can be explained by careful evaluation described here, and unexplained incomplete penetrance is very rare.

The first line of pLoF assessment in this study was based on our framework for loss-of-function curation[21], which evaluates if there is reason to believe the pLoF variants do not result in ablated protein expression. The verdict "Not LoF/Likely not LoF" does not determine variant pathogenicity, e.g., a 3' pLoF variant not resulting in lost protein expression can still be pathogenic by expression of a truncated protein. Of the 734 pLoF variants evaluated with this framework, 108 were also reported as B, LB, VUS, LP or P in ClinVar which allowed us to demonstrate that indeed the majority of explained variants in the last exon, low pext regions, or part of multinucleotide variants do not result in LoF and are benign, and only under rare circumstances P/LP. Pathogenicity must be assessed considering specific gene properties, such as the presence of a functional domain or nearby established pathogenic variants. Further, a lowly expressed region (low pext score region) that shows evidence of biological relevance, e.g., expression in

a disease-relevant tissue, can still have important implications in disease. The current restriction to adult tissues in GTEx, and lack of any pediatric or prenatal tissues, is limiting when interpreting transcript properties and expression of genes causing syndromes manifesting prenatally.

Somatic occurrence and clonal expansion over time likely explain some germline-resembling unexpected variants. For one, we observed an overrepresentation of pLoF variants in severe pediatric disease genes in samples from the 1000 Genomes project, and to a lesser extent, a general enrichment for (germline blood) samples originating from cancer cohorts. Samples from the 1000 Genomes project are derived from cultured cell lines, suggesting that these variants may be somatic variants that have gone through clonal expansion over time in cell culture, rather than germline in origin. This highlights the need for caution when using non-primary cells to study the landscape of human variation tolerance and is one of the reasons that the 1000 Genomes Project (1KG) allele information is available on each variant page as a separate tab of the gnomAD v4 frequencies table, allowing identification of variants originating from this project. Second, we observed skewed age distribution towards elderly individuals of samples with variants in clonal hematopoiesis genes, as well as variants labeled because of low alternative AB, demonstrating how skewed age can help guide the identification of somatic occurrence of variants, as shown previously[28].

We found that 9 of 734 pLoF variants fell in exons that have been associated with alternative splicing mediated by specific sQTL variants. sQTLs have previously been suggested to play a role in penetrance. Einson et al., showed that natural selection acts on haplotype configurations that reduce the transcript inclusion of putatively pathogenic variants in TOPMed consortium data, especially when limiting to haploinsufficient genes[17], and Beaumont et al., reported a general observation of the non-uniform distribution of pLoF variants in haploinsufficienct disease-associated genes and suggested that alternative splicing and translation re-initiation of these regions could result in incomplete penetrance[34]. We present a specific example of an sQTL variant associated with alternative splicing of a certain region of *MEF2C*. *MEF2C* haploinsufficiency is associated with a neurodevelopmental disorder presenting with developmental and cognitive delay, limited language and walking, hypotonia, and seizures (MIM: 600662)[35]. The combination of clustering of pLoF variants gnomAD (fourteen pLoF) and ClinVar P/LP variants (three P/LP and four VUS pLoF) could suggest that pLoF in this region can be of incomplete penetrance in the gnomAD individuals, potentially by alternative splicing that could be mediated by the sQTL carried by three out of four individuals with sQTL genotype data and pLoF variants in this region. It is also possible that the alternative splicing is population-wide, pLoF in this region is not associated with disease, and the P/LP variants in this region are examples of misclassifications in ClinVar. Either way, in these cases of alternative splicing, a lowered pext score is a powerful tool to highlight regions of less biological relevance.

This study helps further our understanding and ability to interpret variant effects and understand incomplete penetrance of disease, especially focusing on pLoF variants. Most importantly, we highlight the complicated assessment of pLoF variants effect, which is a major challenge in variant classification, interpretation, diagnostic testing, and genetic risk prediction. Most presumed disease-causing pLoF variants here and in other population databases can be reclassified as misannotations, somatic, or artifacts by deep investigation, but also include cases of incomplete penetrance mediated by other modifying genetic variants. Only 4.5% of pLoF variants associated with the 77 high-penetrant conditions were unexplained, highlighting that deep assessment on a variant-by-variant basis far beyond standard high-throughput pipelines is useful and needed. Failing to do so runs the risk of overinterpreting pathogenicity in clinic and in research,

especially for severe conditions in unaffected individuals where the false discovery rate is higher[36].

Although not within the scope of this study, molecular assessment of these unexpected cases will be useful next steps and allow a deeper understanding of biological mechanisms underlying disease-penetrance. Continued aggregation of sequencing data, ideally associated with phenotype, will allow further studies in this area. Especially when focusing on under-represented groups where yet undiscovered haplotypes with modifying events, combined with known disease variants, can inform new mechanisms resulting in incomplete penetrance of disease.

## Methods

### Ethical approval
Individuals in gnomAD have consented to research and we have complied with all relevant ethical regulations. The Broad Institute of MIT and Harvard Office of Research Subject Protection, and Mass General Brigham IRB approved this work.

### ClinVar variants in gnomAD
ClinVar variants reported here include all variants available for download by December 1st, 2023 (https://ftp.ncbi.nlm.nih.gov/pub/clinvar/tab_delimited/) filtered to indels less than 50 base pairs and single nucleotide variants on chromosomes 1:22 + X + Y. All variants were grouped based on the aggregate ClinVar classification category (clinical significance) into B/LB when reported as "Likely benign", "Benign" or "Benign/Likely benign", Uncertain when "Uncertain significance", P/LP when "Pathogenic", "Likely pathogenic" or "Pathogenic/Likely pathogenic" and conflicting when "Conflicting interpretations of pathogenicity". A small proportion of variants that did not fall into any of the above-listed classification categories were excluded; in the majority of these the clinical significance had not been provided ("not provided"/ "no interpretation for the single variant") followed by "drug response", "risk factor" and "association". For "ClinVar variants present in gnomAD" we included any ClinVar variant represented in the 807,162 individuals that passed gnomAD QC filters[20], only including a variant if "pass" in either exomes or genomes. A combined AF was calculated for variants detected with both exome and genome sequencing. For variants only identified by one sequencing method the AF of samples sequenced by that method was used. The inheritance pattern for each pathogenic variant was determined using the reported inheritance in OMIM of the relevant gene (by October 23, 2023), categorized as AR, AR/AD (if both patterns were reported), AD or XL. Variants that fell in genes not reported in OMIM, genes with no reported inheritance, or other types/combinations of inheritance were excluded from the inheritance analysis.

### Rescue by local pLoF events in a subset of P/LP in ClinVar
We investigated if a set of P/LP variants were tolerated due to modifying pLoF variants in the same gene, potentially reducing the expression of the pathogenic allele. We included all P/LP ClinVar variants found in fewer than 50 of 807,162 individuals in gnomAD that were located in a gene not constrained for LoF (predicted loss-of-function intolerance < 0.9, pLI) and with at least one condition of AD inheritance in OMIM, suggesting that pathogenicity is more likely to act through a gain-of-function mechanism. The pLI score is mostly stable over v2 and v4 versions and its dichotomous nature makes it suitable for determining haploinsufficiency in the context of variant depletion in a population database. Of note, the Loss-of-function Observed/Expected Upper-bound Fraction (LOEUF) score, a continuous metric of pLoF depletion, shows some variance between versions due to sample size increases, resulting in an increased discovery rate of ultra-rare variants along with an increased number of artifacts (Figure S10). We then manually assessed all pairs of P/LP variants occurring in combination with a pLoF event with a global AF equal or less than 1% (AF ≤ 0.01) that passed gnomAD QC filters (depth < 10, genotype quality < 20, minor AB < 0.2 for alternate alleles of heterozygous genotypes). Combinations were determined interesting if passing three criteria (1) carriership of a pLoF variant in more than 50% of individuals with the P/LP variant (2) a pLoF variant determined as a true LoF resulting in ablated protein product using LoF curation explained below (filtering artifacts, somatic variants, missanotations or rescued variant), (3) a P/LP variant acting through dominant gain-of-function mechanism.

### Severe disease genes investigated in gnomAD
We assessed over 450 genes associated with rare disorders starting from gene lists from resources like the Deciphering Developmental Disorders (DDD) study and the ClinGen Dosage sensitivity curation project as well as internally collected genes and genes shared by collaborators. By literature review, we filtered the >450 genes to a stringent gene-disease list where all genes were associated with disorders that met the following criteria: (1) Caused by autosomal haploinsufficiency in at least three unrelated cases. (2) Early-onset, defined as before the age of three. (3) Severe phenotype unlikely to be compatible with participation in common disease studies or prohibit consent to such study, i.e., mainly inclusion of syndromes resulting in severe congenital malformations and/or neurodevelopmental symptoms of severe degree. (4) Highly penetrant and limited variable expressivity of phenotypes, scored as estimated penetrance > 70%, > 80%, > 90% or 100%. For many disorders there is no clear penetrance estimate available in literature or within resources like GeneReviews. For these disorders, the reported proportion of de novo versus inherited variants was used as a proxy for penetrance.

### Loss-of-function curation and splicing quantitative trait loci analysis
We included all pLoF variants (nonsense, frameshift, essential splice variants) found in v4.1.0 genomes (76,215 individuals) in any GENCODE 39 transcripts that were high-confidence according to the Loss-Of-Function Transcript Effect Estimator[4]. LoF variants were assessed using the framework for LoF curation previously published by this group[21], adapted for this project. In general, we used conservative thresholds to exclude any pLoF variants likely not to cause loss of function. The specific set of rules used for this project are found in Supplementary Data S5. Splice variants were assessed for rescues (within 1000 base pairs) using SpliceAI lookup (https://spliceailookup.broadinstitute.org/). pLoF variants labeled as Not LoF or Likely not LoF were grouped and categorized as "Explained" (blue), variants labeled as Uncertain were excluded (gray), and variants scored as LoF or Likely LoF were grouped and referred to as "unexplained" (red).

Splicing quantitative trait loci (sQTL) analysis was performed on pLoF variants in v4 genomes according to methods previously described[17]. In short, cis-sQTL variants were identified from GTEx v8 data by association between exon inclusion levels (PSI) and genetic variants in 1 Mb window, at < 5% FDR.

### Statistical analysis and data visualization
All statistical analysis and data visualization for figures were generated using R v4.3.1 (https://www.r-project.org/), mainly utilizing libraries from ggplot2 v3.4.3, ComplexUpset v1.3.3, and ggpubr v0.6.0.

### Reporting summary
Further information on research design is available in the Nature Portfolio Reporting Summary linked to this article.

## Data availability
Variant data, constraint metrics for gnomAD v4 and v2, and loss of function curation data are publicly available at https://gnomad.broadinstitute.org/data.

## Code availability

Code used for analysis on publicly available gnomAD data is available at GitHub (DOI: 10.5281/zenodo.15175046). The code to generate figures can be shared upon request.

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

## Acknowledgements

We thank the individuals and researchers whose data is in gnomAD for their contributions. We thank Dr. Mark Daly, Sinéad Chapman, Caroline Cusick, and the team for sharing access to sequencing read data for the *GJB2* case. S.G. was supported by the Knut and Alice Wallenberg Foundation scholarship program for postdoctoral studies at the Broad Institute. S.L.S. was supported by a Manton Center for Orphan Disease Research Fellowship at Boston Children's Hospital. D.G.M. was supported by a National Health and Medical Research Council (Australia) investigator grant 2009982. The study was funded in part by the Harold and Leila Y. Mathers Charitable Foundation to S.G. and A.O.D.L. and by the NIH/NHGRI grant U24HG011450 to H.L.R. Extended information about grants supporting the Genome Aggregation Database Consortium is found in Supplementary Information.

## Author contributions

S.G. and A.O.D.L. developed the project. S.G. planned and conducted all analyses, wrote the manuscript, and prepared all figures and Supplementary Data. M.S.B. and N.A.W. contributed to LoF curation. S.L.S., J.K.G., and M.W. contributed to analyses on gnomAD data. J.E. and T.L. performed the sQTL analysis. H.L.R., D.G.M., A.O.D.L. contributed with study design, funding, and feedback. All authors approved the final version of the manuscript.

## Competing interests

A.O.D.L. is on the scientific advisory board for Congenica, receives research funding in the form of reagents from Pacific Biosciences, and a paid advisor to Addition Therapeutics and former paid advisor to Tome Biosciences, Addition Therapeutics, and Ono Pharma USA. D.G.M. is a paid adviser to GlaxoSmithKline, Insitro, and Overtone Therapeutics, and receives research funding from Microsoft Corporation. H.L.R. has received rare disease research funding from Microsoft and Illumina and compensation as a past member of the scientific advisory board of Genome Medical. T.L. is an advisor and has equity in Variant Bio. The remaining authors declare no competing interests. Extended information about competing interests for the Genome Aggregation Database Consortium is found in Supplementary Information.

## Additional information

## Genome Aggregation Database Consortium

Maria Abreu[8], Amina Abubakar[9], Rolf Adolfsson[10,11], Carlos A. Aguilar Salinas[12], Tariq Ahmad[13], Christine M. Albert[14,15], Jessica Alföldi[1,2], Matthieu Allez[16], Celso Arango López[17,18,19], Diego Ardissino[20], Irina M. Armean[1,2], Elizabeth G. Atkinson[21,22], Gil Atzmon[23,24], Eric Banks[25], John Barnard[26], Samantha M. Baxter[1], Laurent Beaugerie[27], David Benjamin[25], Emelia J. Benjamin[28,29,30], Louis Bergelson[25], Charles Bernstein[31,32], Douglas Blackwood[33], Michael Boehnke[34], Lori L. Bonnycastle[35], Erwin P. Bottinger[36], Donald W. Bowden[37], Matthew J. Bown[38,39], Harrison Brand[40,41], Steven Brant[42,43,44], Ted Brookings[25,45], Sam Bryant[2,22], Shawneequa L. Callier[46,47], Hannia Campos[48,49], John C. Chambers[50,51,52], Juliana C. Chan[53], Katherine R. Chao[1,2], Sinéad Chapman[1,2,22], Daniel I. Chasman[14,54], Lea A. Chen[55], Siwei Chen[1,2], Rex Chisholm[56], Judy Cho[36], Rajiv Chowdhury[57], Mina K. Chung[58], Wendy K. Chung[59], Kristian Cibulskis[25], Bruce Cohen[60,61], Ryan L. Collins[1,40,62], Kristen M. Connolly[63], Adolfo Correa[64], Aiden Corvin[65], Miguel Covarrubias[25], Nick Craddock[66], Beryl B. Cummings[1,62], Dana Dabelea[67], Mark J. Daly[1,2,68], John Danesh[57], Dawood Darbar[69], Phil Darnowsky[1], Joshua C. Denny[70], Stacey Donnelly[71], Richard H. Duerr[72,73,74], Ravindranath Duggirala[75], Josée Dupuis[76,77], Patrick T. Ellinor[1,78], Roberto Elosua[79,80,81], James Emery[25], Eleina England[1,82], Jeanette Erdmann[83,84,85], Tõnu Esko[86], Emily Evangelista[1], Yossi Farjoun[87], Diane Fatkin[88,89,90], William Faubion[91], Steven Ferriera[92], Gemma Figtree[93,94,95], Kelly Flannagan[96], Jose Florez[54,97,98], Laurent Francioli[1,2], Andre Franke[99,100], Adam Frankish[101], Jack Fu[1,40,41], Martti Färkkilä[102,103,104], Stacey Gabriel[92], Kiran Garimella[25], Laura D. Gauthier[25], Jeff Gentry[25], Michel Georges[105], Gad Getz[54,106,107], David C. Glahn[108,109], Benjamin Glaser[110], Stephen J. Glatt[111], Fernando S. Goes[112], David Goldstein[113,114], Clicerio Gonzalez[115], Julia Goodrich[1,40], Riley H. Grant[1], Leif Groop[116,117], Sanna Gudmundsson[1,3,4], Namrata Gupta[1,92], Andrea Haessly[25], Christopher Haiman[118], Ira Hall[119], Craig L. Hanis[120], James Hanyok[121], Matthew Harms[122,123], Qin He[1], Mikko Hiltunen[124], Matti M. Holi[125], Christina M. Hultman[126,127], Steve Jahl[1,2], Chaim Jalas[128], Thibault Jeandet[25], Mikko Kallela[129], Diane Kaplan[25], Jaakko Kaprio[117], Konrad J. Karczewski[1,2,22], Elizabeth W. Karlson[130], Sekar Kathiresan[40,54,131], Eimear E. Kenny[132], Bong-Jo Kim[133], Young Jin Kim[133], Daniel King[1], George Kirov[134], Zan Koenig[2,22], Jaspal Kooner[51,135,136], Seppo Koskinen[137], Harlan M. Krumholz[138,139], Subra Kugathasan[140], Juozas Kupcinskas[141], Soo Heon Kwak[142], Markku Laakso[143,144], Nicole Lake[145], Mikael Landén[126,146], Trevyn Langsford[25], Kristen M. Laricchia[1,2], Terho Lehtimäki[147,148], Monkol Lek[145], James Lewis[149,150,151], Cecilia M. Lindgren[152,153,154], Emily Lipscomb[1], Christopher Llanwarne[25], Ruth J. F. Loos[36,155],

Edouard Louis[156], Chelsea Lowther[40], Wenhan Lu[1], Steven A. Lubitz[1,78], Tom Lyons[157], Ronald C. W. Ma[53,158,159], Daniel G. MacArthur[6,7], Dara S. Manoach[160,161], Gregory M. Marcus[162], Jaume Marrugat[163,164], Nicholas Marston[165,166], Daniel M. Marten[1,3], Alicia R. Martin[1,2,22], Kari M. Mattila[167], Steven McCarroll[22,168], Mark I. McCarthy[169,170,171], Jacob L. McCauley[172,173], Dermot McGovern[174], Ruth McPherson[175], Andrew MacQuillin[176], James B. Meigs[1,54,177], Olle Melander[178], Andres Metspalu[86], Deborah Meyers[179], Eric V. Minikel[1], Braxton D. Mitchell[180], Paul Moayyedi[181,182,183], Sanghamitra Mohanty[184], Andrés Moreno Estrada[185], Nicola J. Mulder[186,187], Ruchi Munshi[25], Aliya Naheed[188], Andrea Natale[189,190,191], Saman Nazarian[192,193], Benjamin M. Neale[1,2], Charles Newton[194,195], Peter M. Nilsson[196], Sam Novod[25], Anne H. O'Donnell-Luria[1,3,40], Michael C. O'Donovan[66], Yukinori Okada[197,198,199], Dost Ongur[54,60], Roel Ophoff[200,201,202], Lorena Orozco[203,204], Willem Ouwehand[205], Michael J. Owen[66], Nick Owen[206], Colin Palmer[207], Nicholette D. Palmer[37], Aarno Palotie[2,22,117], Mara Parellada[17,208,209], Kyong Soo Park[142,210], Carlos Pato[211], Nancy L. Pedersen[126], Tina Pesaran[212], Nikelle Petrillo[25], William Phu[1,3], Sharon Plon[213], Danielle Posthuma[214,215], Timothy Poterba[1,2,22], Ann E. Pulver[112], Aaron Quinlan[216], Dan Rader[192,217], Nazneen Rahman[218], Heidi Rehm[1,40], Andreas Reif[219], Alex Reiner[220,221], Anne M. Remes[222,223], Dan Rhodes[1], Stephen Rich[224,225], John D. Rioux[226,227], Samuli Ripatti[71,117,228], David Roazen[25], Jason Roberts[229,230], Elise Robinson[40], Dan M. Roden[231,232], Jerome I. Rotter[233], Guy Rouleau[234], Valentin Ruano-Rubio[25], Christian T. Ruff[235,236,237], Heiko Runz[238], Marc S. Sabatine[239,240,241], Nareh Sahakian[25], Danish Saleheen[242,243,244], Veikko Salomaa[245], Andrea Saltzman[1], Nilesh J. Samani[39,246], Kaitlin E. Samocha[1,40], Alba Sanchis-Juan[40], Akira Sawa[247,248,249], Jeremiah Scharf[1,22,40], Molly Schleicher[1], Patrick Schultz[22], Heribert Schunkert[250,251], Sebastian Schönherr[252], Eleanor G. Seaby[1,253], Cotton Seed[2,22], Svati H. Shah[254,255], Megan Shand[25], Ted Sharpe[25], Moore B. Shoemaker[256], Tai Shyong[257,258], Edwin K. Silverman[240,259], Moriel Singer-Berk[1], Jurgita Skieceviciene[141], Pamela Sklar[260,261,262], J. Gustav Smith[263,264,265], Jonathan T. Smith[25], Jordan Smoller[40], Hilkka Soininen[266], Harry Sokol[267,268,269], Matthew Solomonson[1,2], Rachel G. Son[1], Jose Soto[25], Tim Spector[270], David St Clair[271], Christine Stevens[1,2,22], Nathan O. Stitziel[272,273,274], Patrick F. Sullivan[126,275], Jaana Suvisaari[245], E. Shyong Tai[276,277,278], Michael E. Talkowski[1,22,40], Yekaterina Tarasova[1], Kent D. Taylor[233], Yik Ying Teo[276,279,280], Grace Tiao[1,2], Kathleen Tibbetts[25], Charlotte Tolonen[25], Ming Tsuang[281,282], Tiinamaija Tuomi[117,283,284], Dan Turner[285], Teresa Tusie-Luna[286,287], Erkki Vartiainen[228], Marquis Vawter[288], Severine Vermeire[289,290], Elisabet Vilella[291,292,293], Christopher Vittal[1,2], Gordon Wade[25], Mark Walker[25], Arcturus Wang[1,2,22], Lily Wang[1,294], Qingbo Wang[1,197], James S. Ware[1,295,296], Hugh Watkins[297], Nicholas A. Watts[1,2], Rinse K. Weersma[298], Ben Weisburd[25], Maija Wessman[117,299], Christopher Whelan[1], Nicola Whiffin[1,300,301], James G. Wilson[302], Michael W. Wilson[1], Lauren Witzgall[1], Ramnik J. Xavier[303,304], Mary T. Yohannes[1], Robert Yolken[305] & Xuefang Zhao[1]

[8]University of Miami Miller School of Medicine, Gastroenterology, Miami, USA. [9]Neuroscience Research Group, Department of Clinical Sciences, Kenyan Medical Research Institute, Wellcome Trust, Kilifi, Kenya. [10]Department of Clinical Sciences, Psychiatry, Umeå University, Umeå, Sweden. [11]Institute for Human Development, Aga Khan University, Nairobi, Kenya. [12]Unidad de Investigacion de Enfermedades Metabolicas, Instituto Nacional de Ciencias Medicas y Nutricion, Mexico City, Mexico. [13]Peninsula College of Medicine and Dentistry, Exeter, UK. [14]Division of Preventive Medicine, Brigham and Women's Hospital, Boston, MA, USA. [15]Division of Cardiovascular Medicine, Brigham and Women's Hospital and Harvard Medical School, Boston, MA, USA. [16]Gastroenterology Department, Hôpital Saint-Louis - APHP, Université Paris Cité, INSERM, U1160 Paris, France. [17]Department of Child and Adolescent Psychiatry, Institute of Psychiatry and Mental Health, Madrid, Spain. [18]Hospital General Universitario Gregorio Marañón, Madrid, Spain. [19]School of Medicine, Complutense University of Madrid, Madrid, Spain. [20]Department of Cardiology University Hospital, Parma, Italy. [21]Department of Molecular and Human Genetics, Baylor College of Medicine, Houston, TX, USA. [22]Stanley Center for Psychiatric Research, The Broad Institute of MIT and Harvard, Cambridge, MA, USA. [23]Department of Biology Faculty of Natural Sciences, University of Haifa, Haifa, Israel. [24]Departments of Medicine and Genetics, Albert Einstein College of Medicine, Bronx, NY, USA. [25]Data Science Platform, Broad Institute of MIT and Harvard, Cambridge, MA, USA. [26]Department of Quantitative Health Sciences, Lerner Research Institute Cleveland Clinic, Cleveland, OH, USA. [27]Sorbonne Université, APHP, Gastroenterology Department Saint Antoine Hospital, Paris, France. [28]NHLBI and Boston University's Framingham Heart Study, Framingham, MA, USA. [29]Department of Medicine, Boston University Chobanian & Avedisian School of Medicine, Boston, MA, USA. [30]Department of Epidemiology, Boston University School of Public Health, Boston, MA, USA. [31]Department of Internal Medicine, Max Rady College of Medicine, University of Manitoba, Winnipeg, Canada. [32]Rady Faculty of Health Sciences, University of Manitoba, Winnipeg, Canada. [33]Anaesthesia and Perioperative Medicine, Division of Surgery and Interventional Science, University College London Hospitals NHS Foundation Trust, University College London, London, UK. [34]Department of Biostatistics and Center for Statistical Genetics, University of Michigan, Ann Arbor, MI, USA. [35]National Human Genome Research Institute, National Institutes of Health Bethesda, Bethesda, MD, USA. [36]The Charles Bronfman Institute for Personalized Medicine, Icahn School of Medicine at Mount Sinai, New York, NY, USA. [37]Department of Biochemistry, Wake Forest School of Medicine, Winston-Salem, NC, USA. [38]Department of Cardiovascular Sciences and NIHR Leicester Biomedical Research Centre, University of Leicester, Leicester, UK. [39]NIHR Leicester Biomedical Research Centre, Glenfield Hospital, Leicester, UK. [40]Center for Genomic Medicine, Massachusetts General Hospital, Boston, MA, USA. [41]Department of Neurology, Massachusetts General Hospital and Harvard Medical School, Boston, MA, USA. [42]Department of Medicine, Rutgers Robert Wood Johnson Medical School, Rutgers, The State University of New Jersey, New Brunswick, Brunswick, NJ, USA. [43]Department of Genetics and the Human Genetics Institute of New Jersey, School of Arts and Sciences, Rutgers, The State University of New Jersey, Piscataway, NJ, USA. [44]Meyerhoff Inflammatory Bowel Disease Center, Johns Hopkins University School of Medicine, Baltimore, MD, USA. [45]Fulcrum Genomics, Boulder, CO, USA. [46]Department of Clinical Research and Leadership, George Washington University School of Medicine and Health Sciences, Washington, DC, USA. [47]Center for Research on Genomics and Global Health, National Human Genome Research Institute, National Institutes of Health, Bethesda, MD, USA. [48]Harvard School of Public Health, Boston, MA, USA. [49]Central American Population Center, San Pedro, Costa Rica. [50]Department of Epidemiology and Biostatistics, Imperial College London, London, UK. [51]Department of

Cardiology, Ealing Hospital, NHS Trust, Southall, UK. [52]Imperial College, Healthcare NHS Trust Imperial College London, London, UK. [53]Department of Medicine and Therapeutics, The Chinese University of Hong Kong, Hong Kong, China. [54]Department of Medicine, Harvard Medical School, Boston, MA, USA. [55]Department of Medicine, Rutgers Robert Wood Johnson Medical School, New Brunswick, NJ, USA. [56]Northwestern University, Evanston, IL, USA. [57]University of Cambridge, Cambridge, England. [58]Departments of Cardiovascular, Medicine Cellular and Molecular Medicine Molecular Cardiology, Quantitative Health Sciences, Cleveland Clinic, Cleveland, OH, USA. [59]Department of Pediatrics, Boston Children's Hospital, Harvard Medical School, Boston, MA, USA. [60]McLean Hospital, Belmont, MA, USA. [61]Department of Psychiatry, Harvard Medical School, Boston, MA, USA. [62]Division of Medical Sciences, Harvard Medical School, Boston, MA, USA. [63]Genomics Platform, Broad Institute of MIT and Harvard, Cambridge, MA, USA. [64]Department of Medicine, University of Mississippi Medical Center, Jackson, MI, USA. [65]Neuropsychiatric Genetics Research Group, Dept of Psychiatry and Trinity Translational Medicine Institute, Trinity College Dublin, Dublin, Ireland. [66]Centre for Neuropsychiatric Genetics & Genomics, Cardiff University School of Medicine, Cardiff, Wales, UK. [67]Department of Epidemiology Colorado School of Public Health Aurora, Aurora, CO, USA. [68]Institute for Molecular Medicine Finland, (FIMM), Helsinki, Finland. [69]Department of Medicine and Pharmacology, University of Illinois at Chicago, Chicago, IL, USA. [70]Vanderbilt University Medical Center, Nashville, TN, USA. [71]Broad Institute of MIT and Harvard, Cambridge, MA, USA. [72]Department of Medicine, School of Medicine, University of Pittsburgh, Pittsburgh, PA, USA. [73]Department of Human Genetics, School of Public Health, University of Pittsburgh, Pittsburgh, PA, USA. [74]Clinical and Translational Science Institute, University of Pittsburgh, Pittsburgh, PA, USA. [75]Department of Life Sciences, College of Arts and Scienecs, Texas A&M University-San Antonio, San Antonio, TX, USA. [76]Department of Biostatistics, Boston University School of Public Health, Boston, MA, USA. [77]Department of Epidemiology, Biostatistics and Occupational Health, McGill University, Montreal, QC, Canada. [78]Cardiac Arrhythmia Service and Cardiovascular Research Center, Massachusetts General Hospital, Boston, MA, USA. [79]Cardiovascular Epidemiology and Genetics, Hospital del Mar Medical Research Institute (IMIM), Barcelona, Catalonia, Spain. [80]CIBER CV, Barcelona, Spain. [81]Departament of Medicine, Faculty of Medicine, University of Vic-Central University of Catalonia, Vic Catalonia, Spain. [82]Clalit Genomics Center, Ramat-Gan, Israel. [83]Institute for Cardiogenetics, University of Lübeck, Lübeck, Germany. [84]German Research Centre for Cardiovascular Research, Hamburg/Lübeck/Kiel, Lübeck, Germany. [85]University Heart Center Lübeck, Lübeck, Germany. [86]Estonian Genome Center, Institute of Genomics University of Tartu, Tartu, Estonia. [87]Richards Lab, Lady Davis Institute, Montreal, QC, Canada. [88]Victor Chang Cardiac Research Institute, Darlinghurst, NSW, Australia. [89]Faculty of Medicine and Health, UNSW Sydney, Kensington, NSW, Australia. [90]Cardiology Department, St Vincent's Hospital, Darlinghurst, NSW, Australia. [91]Mayo Clinic, Arizona, USA. [92]Broad Genomics, Broad Institute of MIT and Harvard, Cambridge, MA, USA. [93]Cardiovascular Discovery Group, Kolling Institute of Medical Research, University of Sydney, Sydney, Australia. [94]Department of Cardiology, Royal North Shore Hospital, Sydney, Australia. [95]Faculty of Medicine and Health, University of Sydney, Sydney, Australia. [96]OSRP, Broad Institute of MIT and Harvard, Cambridge, MA, USA. [97]Diabetes Unit (Department of Medicine) and Center for Genomic Medicine, Massachusetts General Hospital, Boston, MA, USA. [98]Programs in Metabolism and Medical & Population Genetics, Broad Institute of MIT and Harvard, Cambridge, MA, USA. [99]Institute of Clinical Molecular Biology, Christian-Albrechts-University of Kiel, Kiel, Germany. [100]University Hospital Schleswig-Holstein, Kiel, Germany. [101]European Molecular Biology Laboratory, European Bioinformatics Institute, Wellcome Genome Campus, Hinxton, UK. [102]Helsinki University and Helsinki University Hospital Clinic of Gastroenterology, Helsinki, Finland. [103]Helsinki University and Helsinki University Hospital, Helsinki, Finland. [104]Abdominal Center, Helsinki, Finland. [105]Unit of Animal Genomics, GIGA & Faculty of Veterinary Medicine, University of Liège, Liège, Belgium. [106]Bioinformatics Program MGH Cancer Center and Department of Pathology, Boston, MA, USA. [107]Cancer Genome Computational Analysis, Broad Institute of MIT and Harvard, Cambridge, MA, USA. [108]Department of Psychiatry and Behavioral Sciences, Boston Children's Hospitaland Harvard Medical School, Boston, MA, USA. [109]Harvard Medical School Teaching Hospital, Boston, MA, USA. [110]Department of Endocrinology and Metabolism, Hadassah Medical Center and Faculty of Medicine, Hebrew University of Jerusalem, Jerusalem, Israel. [111]Department of Psychiatry and Behavioral Sciences, SUNY Upstate Medical University, Syracuse, NY, USA. [112]Department of Psychiatry and Behavioral Sciences, Johns Hopkins University School of Medicine, Baltimore, MD, USA. [113]Institute for Genomic Medicine, Columbia University Medical Center Hammer Health Sciences, New York, NY, USA. [114]Department of Genetics & Development Columbia University Medical Center, Hammer Health Sciences, New York, NY, USA. [115]Centro de Investigacion en Salud Poblacional, Instituto Nacional de Salud Publica, Publica, Mexico. [116]Lund University Sweden, Lund, Sweden. [117]Institute for Molecular Medicine Finland, (FIMM) HiLIFE University of Helsinki, Helsinki, Finland. [118]Center for Genetic Epidemiology, Department of Population and Public Health Sciences, University of Southern California, Los Angeles, CA, USA. [119]Washington School of Medicine, St Louis, MI, USA. [120]Human Genetics Center, University of Texas Health Science Center at Houston, Houston, TX, USA. [121]Daiichi Sankyo, Basking Ridge, NJ, USA. [122]Department of Neurology Columbia University, New York City, New York, NY, USA. [123]Institute of Genomic Medicine, Columbia University, New York City, New York, NY, USA. [124]Institute of Biomedicine, University of Eastern Finland, Kuopio, Finland. [125]Department of Psychiatry, Helsinki University Central Hospital Lapinlahdentie, Helsinki, Finland. [126]Department of Medical Epidemiology and Biostatistics, Karolinska Institutet, Stockholm, Sweden. [127]Icahn School of Medicine at Mount Sinai, New York, NY, USA. [128]Bonei Olam, Center for Rare Jewish Genetic Diseases, Brooklyn, NY, USA. [129]Department of Neurology, Helsinki University, Central Hospital, Helsinki, Finland. [130]Division of Rheumatology, Inflammation, and Immunity, Department of Medicine, Brigham and Women's Hospital and Harvard Medical School, Boston, MA, USA. [131]Cardiovascular Disease Initiative and Program in Medical and Population Genetics, Broad Institute of MIT and Harvard, Cambridge, MA, USA. [132]Institute for Genomic Health, Icahn School of Medicine at Mount Sinai, New York, NY, USA. [133]Division of Genome Science, Department of Precision Medicine, National Institute of Health, Seoul, Republic of Korea. [134]MRC Centre for Neuropsychiatric Genetics & Genomics, Cardiff University School of Medicine, Cardiff, Wales, UK. [135]Imperial College Healthcare NHS Trust, London, UK. [136]National Heart and Lung Institute Cardiovascular Sciences, Hammersmith Campus, Imperial College London, London, UK. [137]Department of Health THL-National Institute for Health and Welfare, Helsinki, Finland. [138]Section of Cardiovascular Medicine, Department of Internal Medicine, Yale School of Medicine, New Haven, Connecticut, USA. [139]Center for Outcomes Research and Evaluation, Yale-New Haven Hospital, New Haven, Connecticut, USA. [140]Division of Pediatric Gastroenterology, Emory University School of Medicine, Atlanta, GA, USA. [141]Institute for Digestive Research and Department of Gastroenterology, Lithuanian University of Health Sciences, Kaunas, Lithuania. [142]Department of Internal Medicine, Seoul National University Hospital, Seoul, Republic of Korea. [143]The University of Eastern Finland, Institute of Clinical Medicine, Kuopio, Finland. [144]Kuopio University Hospital, Kuopio, Finland. [145]Department of Genetics, Yale School of Medicine, New Haven, CT, USA. [146]Department of Neuroscience and Physiology, University of Gothenburg, Gothenburg, Sweden. [147]Department of Clinical Chemistry Fimlab Laboratories, Tampere University, Tampere, Finland. [148]Finnish Cardiovascular Research Center-Tampere Faculty of Medicine and Health Technology, Tampere University, Tampere, Finland. [149]State Key Laboratory of Coal Conversion, Institute of Coal Chemistry, Chinese Academy of Sciences, Taiyuan 030001, China. [150]National Energy Center for Coal to Liquids, Synfuels China Company, Ltd., Huairou District, Beijing 101400, China. [151]Hong Kong Quantum AI Laboratory, Ltd., Hong Kong Science Park, Hong Kong 999077, China. [152]Big Data Institute, Li Ka Shing Centre for Health Information and Discovery, University of Oxford, Oxford, UK. [153]Wellcome Trust Centre Human Genetics, University of Oxford, Oxford, UK. [154]Medical and Population Genetics Program, Broad Institute of MIT and Harvard, Cambridge, MA, USA. [155]The Novo Nordisk Foundation Center for Basic Metabolic Research, Faculty of Health and Medical Sciences, University of Copenhagen, Copenhagen, Denmark. [156]Department of Gastroenterology, University Hospital CHU of Liège, Liège, Belgium. [157]The Department of Health, Alcohol and Other Drugs Strategy Team, Victorian State Government, Melbourne, Victoria, Australia. [158]Li Ka Shing Institute of Health

Sciences, The Chinese University of Hong Kong, Hong Kong, China. [159]Hong Kong Institute of Diabetes and Obesity, The Chinese University of Hong Kong, Hong Kong, China. [160]Department of Psychiatry, Massachusetts General Hospital, Boston, MA, USA. [161]Division of Sleep Medicine, Harvard Medical School, Boston, MA, USA. [162]Division of Cardiology, University of California San Francisco, San Francisco, CA, USA. [163]Hospital del Mar Medical Research Institute (IMIM), Barcelona, Spain. [164]CIBERCV, Madrid, Spain. [165]Division of Cardiovascular Medicine, Brigham and Women's Hospital, Harvard Medical School, Boston, MA, USA. [166]TIMI Study Group, Division of Cardiovascular Medicine (N.M.), Harvard Medical School, Boston, MA, USA. [167]Department of Clinical Chemistry Fimlab Laboratories and Finnish Cardiovascular Research Center-Tampere Faculty of Medicine and Health Technology, Tampere University, Tampere, Finland. [168]Department of Genetics, Harvard Medical School, Boston, MA, USA. [169]Oxford Centre for Diabetes, Endocrinology and Metabolism, University of Oxford, Churchill Hospital Old Road Headington, Oxford, OX, LJ, UK. [170]Welcome Centre for Human Genetics, University of Oxford, Oxford, OX, BN, UK. [171]Oxford NIHR Biomedical Research Centre, Oxford University Hospitals, NHS Foundation Trust, John Radcliffe Hospital, Oxford, OX, DU, UK. [172]John P. Hussman Institute for Human Genomics, Leonard M. Miller School of Medicine, University of Miami, Miami, FL, USA. [173]The Dr. John T. Macdonald Foundation Department of Human Genetics, Leonard M. Miller School of Medicine, University of Miami, Miami, FL, USA. [174]F. Widjaja Foundation Inflammatory Bowel and Immunobiology Research Institute Cedars-Sinai Medical Center, Los Angeles, CA, USA. [175]Atherogenomics Laboratory University of Ottawa, Heart Institute, Ottawa, Canada. [176]Division of Psychiatry, University College London, London, UK. [177]Division of General Internal Medicine, Massachusetts General Hospital, Boston, MA, USA. [178]Department of Clinical Sciences University, Hospital Malmo Clinical Research Center, Lund University, Malmö, Sweden. [179]University of Arizona Health Science, Tuscon, AZ, USA. [180]University of Maryland School of Medicine, Baltimore, MD, USA. [181]The Population Health Research Institute, McMaster University and Hamilton Health Sciences, Hamilton, Ontario, Canada. [182]Division of Gastroenterology, Department of Medicine, McMaster University, Hamilton, Ontario, Canada. [183]Farncombe Family Digestive Health Research Institute, McMaster University, Hamilton, Ontario, Canada. [184]Department of Electrophysiology, Texas Cardiac Arrhythmia Institute, St. David's Medical Center, Austin, Texas, USA. [185]National Laboratory of Genomics for Biodiversity (UGA-LANGEBIO), Irapuato, Mexico. [186]Computational Biology Division, Department of Integrative Biomedical Sciences, University of Cape Town, Cape Town, South Africa. [187]Institute of Infectious Disease & Molecular Medicine, Faculty of Health Sciences, University of Cape Town, Cape Town, South Africa. [188]International Centre for Diarrhoeal Disease Research, Dhaka, Bangladesh. [189]Texas Cardiac Arrhythmia Institute, St. David's Medical Center, Austin, TX, USA. [190]Interventional Electrophysiology, Scripps Clinic, La Jolla, CA, USA. [191]MetroHealth Medical Center, Case Western Reserve University School of Medicine, Cleveland, OH, USA. [192]Perelman School of Medicine, University of Pennsylvania, Philadelphia, PA, USA. [193]Johns Hopkins Bloomberg School of Public Health, Baltimore, MD, USA. [194]Kenya Medical Research Institute-Wellcome Trust Collaborative Programme, Kilifi, Kenya. [195]Dept of Psychiatry, University of Oxford, Oxford, United Kingdom. [196]Lund University, Dept. Clinical Sciences, Skåne University Hospital, Malmö, Sweden. [197]Department of Statistical Genetics, Osaka University Graduate School of Medicine, Suita, Japan. [198]Laboratory of Statistical Immunology, Immunology Frontier Research Center (WPI-IFReC), Osaka University, Suita, Japan. [199]Integrated Frontier Research for Medical Science Division, Institute for Open and Transdisciplinary Research Initiatives, Osaka University, Suita, Japan. [200]Center for Neurobehavioral Genetics, Semel Institute for Neuroscience and Human Behavior, University of California Los Angeles, California, USA. [201]Department of Human Genetics, David Geffen School of Medicine, University of California Los Angeles, Los Angeles, CA, USA. [202]Department of Psychiatry, Erasmus University Medical Center, Rotterdam, The Netherlands. [203]Instituto Nacional de Medicina Genómica, (INMEGEN) Mexico City, Mexico, Mexico. [204]Laboratory of Immunogenomics and Metabolic Diseases, INMEGEN, Mexico City, Mexico, Mexico. [205]Department of Haematology, University of Cambridge, Cambridgeshire, United Kingdom of Great Britain and Northern Ireland, Cambridge, UK. [206]Applied Biomechanics Department, Swansea University, Singleton Park, Swansea SA2 8PP, UK. [207]Medical Research Institute, Ninewells Hospital and Medical School University of Dundee, Dundee, UK. [208]Centro de Investigación Biomédica en Red de Salud Mental (CIBERSAM), Instituto de Salud Carlos III, Madrid, Spain. [209]Hospital General Universitario Gregorio Marañón, School of Medicine, Universidad Complutense, IiSGM, Madrid, Spain. [210]Department of Molecular Medicine and Biopharmaceutical Sciences, Graduate School of Convergence Science and Technology, Seoul National University, Seoul, Republic of Korea. [211]Department of Psychiatry Keck School of Medicine at the University of Southern California, Los Angeles, CA, USA. [212]Ambry Genetics, Aliso Viejo, CA, USA. [213]Department of Pediatrics/Hematology-Oncology, Baylor College of Medicine, Houston, Texas, USA. [214]Department of Complex Trait Genetics, Center for Neurogenomics and Cognitive Research, Amsterdam Neuroscience, Vrije Universiteit Amsterdam, Amsterdam, The Netherlands. [215]Department of Clinical Genetics, Amsterdam Neuroscience, Vrije Universiteit Medical Center, Amsterdam, The Netherlands. [216]Departments of Human Genetics and Biomedical Informatics, University of Utah, Salt Lake City, Utah, UT, USA. [217]Children's Hospital of Philadelphia, Philadelphia, PA, USA. [218]Division of Genetics and Epidemiology, Institute of Cancer Research, London, UK. [219]Department of Psychiatry, Psychosomatic Medicine and Psychotherapy, University Hospital Frankfurt - Goethe University, Frankfurt am Main, Germany. [220]University of Washington, Seattle, WA, USA. [221]Fred Hutchinson Cancer Research Center, Seattle, WA, USA. [222]Medical Research Center, Oulu University Hospital, Oulu, Finland. [223]Research Unit of Clinical Neuroscience Neurology University of Oulu, Oulu, Finland. [224]Center for Public Health Genomics, University of Virginia, Charlottesville, VA, USA. [225]Department of Public Health Sciences, University of Virginia, Charlottesville, VA, USA. [226]Research Center Montreal Heart Institute, Montreal, Quebec, Canada. [227]Department of Medicine, Faculty of Medicine Université de Montréal, Québec, Canada. [228]Department of Public Health Faculty of Medicine, University of Helsinki, Helsinki, Finland. [229]Section of Cardiac Electrophysiology, Division of Cardiology, Department of Medicine, Western University, London, Ontario, Canada. [230]Population Health Research Institute, Hamilton Health Sciences, and McMaster University, Hamilton, Ontario, Canada. [231]Department of Medicine, Vanderbilt, University Medical Center, Nashville, TN, USA. [232]Departments of Pharmacology and Biomedical Informatics Vanderbilt, University Medical Center, Nashville, TN, USA. [233]The Institute for Translational Genomics and Population Sciences, Department of Pediatrics, The Lundquist Institute for Biomedical Innovation at Harbor-UCLA Medical Center, Torrance, CA, USA. [234]Department of Neurology and Neurosurgery, Montreal Neurological Institute and Hospital, McGill University Health Center, Montreal, Canada. [235]TIMI Study Group, Boston, USA. [236]Brigham and Women's Hospital, Boston, USA. [237]Harvard Medical School, Boston, USA. [238]Translational Sciences, Research & Development, Biogen Inc, Cambridge, MA, USA. [239]TIMI Study Group, Division of Cardiovascular Medicine, Brigham and Women's Hospital, Boston, USA. [240]Harvard Medical School, Boston, MA, USA. [241]Ionis Pharmaceuticals, Carlsbad, and the Division of Cardiovascular Medicine, Department of Medicine, University of California, San Diego, La Jolla, USA. [242]Department of Biostatistics and Epidemiology, Perelman School of Medicine, University of Pennsylvania, Philadelphia, PA, USA. [243]Department of Medicine, Perelman School of Medicine at the University of Pennsylvania, Philadelphia, PA, USA. [244]Center for Non-Communicable Diseases, Karachi, Pakistan. [245]National Institute for Health and Welfare, Helsinki, Finland. [246]Department of Cardiovascular Sciences, University of Leicester, Leicester, UK. [247]Departments of Neuroscience, Johns Hopkins University School of Medicine, Baltimore, MD, USA. [248]Departments of Psychiatry, Johns Hopkins University School of Medicine, Baltimore, MD, USA. [249]Departments of Biomedical Engineering, Johns Hopkins University School of Medicine, Baltimore, MD, USA. [250]Department of Cardiology, Deutsches Herzzentrum München, Technical University of Munich, DZHK Munich Heart Alliance, Munich, Germany. [251]Technische Universität München, Munich, Germany. [252]Institute of Genetic Epidemiology, Department of Genetics, Medical University of Innsbruck, 6020 Innsbruck, Austria. [253]Faculty of Medicine, University of Southampton, Southampton SO16 6YD, UK. [254]Duke Molecular Physiology Institute, Durham, NC, USA. [255]Division of Cardiology, Department of Medicine, Duke University School of Medicine, Durham, NC, USA. [256]Division of Cardiovascular Medicine, Nashville VA Medical Center,

Vanderbilt University School of Medicine, Nashville, TN, USA. [257]Division of Endocrinology, National University Hospital, Singapore, Singapore. [258]NUS Saw Swee Hock School of Public Health, Singapore, Singapore. [259]Channing Division of Network Medicine, Brigham and Women's Hospital, Boston, MA, USA. [260]Department of Psychiatry, Icahn School of Medicine at Mount Sinai, New York, NY, USA. [261]Department of Genetics and Genomic Sciences, Icahn School of Medicine at Mount Sinai, New York, NY, USA. [262]Institute for Genomics and Multiscale Biology, Icahn School of Medicine at Mount Sinai, New York, NY, USA. [263]The Wallenberg Laboratory/Department of Molecular and Clinical Medicine, Institute of Medicine, Gothenburg University, Gothenburg, Sweden. [264]Department of Cardiology, Wallenberg Center for Molecular Medicine and Lund University Diabetes Center, Clinical Sciences, Lund University and Skåne University Hospital, Lund, Sweden. [265]Department of Cardiology, Sahlgrenska University Hospital, Gothenburg, Sweden. [266]Institute of Clinical Medicine Neurology, University of Eastern Finad, Kuopio, Finland. [267]Sorbonne Université, INSERM, Centre de Recherche Saint-Antoine, CRSA, AP-HP, Saint Antoine Hospital, Gastroenterology department, F-75012 Paris, France. [268]INRA, UMR1319 Micalis, Jouy en Josas, France. [269]Paris Center for Microbiome Medicine, (PaCeMM) FHU, Paris, France. [270]Department of Twin Research and Genetic Epidemiology King's College London, London, UK. [271]Institute of Medical Sciences, University of Aberdeen, Aberdeen, Scotland, UK. [272]Department of Medicine, Washington University School of Medicine, Saint Louis, MO, USA. [273]Department of Genetics, Washington University School of Medicine, Saint Louis, MO, USA. [274]The McDonnell Genome Institute at Washington University, Saint Louis, MO, USA. [275]Departments of Genetics and Psychiatry, University of North Carolina, Chapel Hill, NC, USA. [276]Saw Swee Hock School of Public Health National University of Singapore, National University Health System, Singapore, Singapore. [277]Department of Medicine, Yong Loo Lin School of Medicine National University of Singapore, Singapore, Singapore. [278]Duke-NUS Graduate Medical School, Singapore, Singapore. [279]Life Sciences Institute, National University of, Singapore, Singapore. [280]Department of Statistics and Applied Probability, National University of, Singapore, Singapore. [281]Center for Behavioral Genomics, Department of Psychiatry, University of California, San Diego, CA, USA. [282]Institute of Genomic Medicine, University of California San Diego, San Diego, CA, USA. [283]Endocrinology, Abdominal Center, Helsinki University Hospital, Helsinki, Finland. [284]Institute of Genetics, Folkhalsan Research Center, Helsinki, Finland. [285]Juliet Keidan Institute of Pediatric Gastroenterology Shaare Zedek Medical Center, The Hebrew University of Jerusalem, Jerusalem, Israel. [286]Instituto de Investigaciones Biomédicas, UNAM, Mexico City, Mexico, Mexico. [287]Instituto Nacional de Ciencias Médicas y Nutrición Salvador Zubirán, Mexico City, Mexico, Mexico. [288]Department of Psychiatry and Human Behavior, University of California Irvine, Irvine, CA, USA. [289]Translational Research in Gastrointestinal Disorders, Department of Chronic Diseases and Metabolism, KU Leuven, Leuven, Belgium. [290]Department of Gastroenterology and Hepatology, University Hospitals Leuven, Leuven, Belgium. [291]Hospital Universitari Institut Pere Mata, Reus, Spain. [292]Institut d'Investigació Sanitària Pere Virgili-CERCA, Tarragona, Spain. [293]Centro de investigación biomédica en red- CIBERSAM, Madrid, Spain. [294]Bioinformatics and Integrative Genomics Program, Harvard Medical School, Boston, MA, USA. [295]National Heart and Lung Institute, Imperial College London, London, UK. [296]UK/MRC Laboratory of Medical Sciences, Imperial College London, London, UK. [297]Radcliffe Department of Medicine, University of Oxford, Oxford, UK. [298]Department of Gastroenterology and Hepatology, University of Groningen and University Medical Center Groningen, Groningen, Netherlands. [299]Folkhälsan Institute of Genetics, Folkhälsan Research Center, Helsinki, Finland. [300]Big Data Institute, University of Oxford, Oxford, UK. [301]Wellcome Centre for Human Genetics, University of Oxford, Oxford, UK. [302]Division of Cardiology, Beth Israel Deaconess Medical Center, Boston, MA, USA. [303]Program in Infectious Disease and Microbiome, Broad Institute of MIT and Harvard, Cambridge, MA, USA. [304]Center for Computational and Integrative Biology, Massachusetts General Hospital, Boston, MA, USA. [305]Stanley Division of Developmental Neurovirology, Department of Pediatrics, Johns Hopkins School of Medicine, Baltimore, MD, USA.

