## [Transparent Peer Review file · Nature Communications]

Exploring penetrance of clinically relevant variants in over 800,000 humans from the Genome Aggregation Database

Corresponding Author: Dr Anne O'Donnell-Luria

Version 0:

Reviewer comments:

Reviewer #1

(Remarks to the Author)

Gudmunsson et al. explore the penetrance of clinically relevant variants in over 800,000 humans from the Genome Aggregation Database. This article addresses a key question for clinical geneticists and more generally for geneticists interested in the genetic architecture of human traits. The paper is very well written, and the methodologies are very relevant. I have no major comments, only suggestions to help the readers.

- For the main part of the study, the authors focused on pLoF, but the first part on the presence of a LoF which suppresses the effect of a missense gain of function variant is highly interesting. It would be interesting to have a dedicated part to the missenses MPC>2. For example, if the identified reasons for incomplete penetrance of LoF also explain the predicted deleterious missense variations. I know that the interpretation of the results for the missenses is more complex than for LoF, but as indicated by the authors most of the VUS are missenses and they represent a burden for the clinical geneticists.

- On page 9, the authors indicate: "We used de novo rate as a proxy for penetrance (Table S4). In these 77 genes, investigating 807,162 individuals, we found 4,464 pLoF variants, of which 3,957 were high-confidence pLoF variants by Loss-Of-Function Transcript Effect Estimator (LOFTEE), 3,223 in exomes (81%), 734 in genomes (19%, Fig. S3)." It is not clear to me if HC-pLoF are more frequent in WES or just that there are more WES data. Sorry for the confusion.

- The authors have chosen to follow up the 734 HC-pLoF of the WGS. I understand that for some analyses (sQTL) only the WGS can be used, but it would be interesting to estimate the proportion of LoF explained and unexplained for the 3,223 HC-pLoF from the WES.

- Page 9, "location in a transcript containing a stop-codon" could be replaced by 'location in a transcript isoform containing a stop-codon'.

- Page 10, the term "evasion" appears ("The observed pLoF evasion rate of 69.8%"). I have an idea of what the authors mean by "evasion" but it would be helpful to have a clear definition.

- The use of the PEXT score is very useful and it would be interesting to have more details on what could have been flagged/filtered out if only brain transcripts were used in the calculation of the PEXT scores. As the authors indicated, there is a lack of prenatal tissue to detect exons specific during fetal life. It would be interesting to test if there are such "prenatal specific exons" in the 77 studied genes and if some of the "explained" pLoF could fall in such exons with low PEXT score in adults and therefore could be miscategorized.

- The discussion is very well done and address many key aspects related to incomplete penetrance.

Figure 2.

- It might be more precise to add introns and exons in the schematic description of a gene. This is even more important when sQTL are presented in the following figures

- I would add a label such as "dominant negative mutation" for the Gly45Glu variation.

Figure 3.

- I would suggest to split the panel D in two. One panel with the pLOF with a single explanation and one allowing a better view of the "combination of explanations". Second, it is not clear to me if "other tx rescue", "SpliceAI rescue" or "SpliceAI no Loss" are independent. They all concern 4.7% of the pLOF but they are annotated as a single explanation?

Reviewer #2

(Remarks to the Author)

Gudmundsson, et al. present a manuscript exploring the frequency of ClinVar variants in gnomAD with the goal of exploring penetrance of disease. 734 pLOF variants in 77 genes associated with severe, early-onset, highly penetrant haploinsufficient disease were present in gnomAD. Analysis showed explanations for the presumed lack of disease manifestation in 701 of the variants (95%). Overall, the authors do support that unexplained incomplete penetrance is very rare. The manuscript is very well written, methods are sound, and support is given for this "exploration" (an accurate and well-used word for this title). This data is very informative and widely applicable to variant analysis in rare disease. While unexplained incomplete penetrance is rare, I would still like to hear more discussion (or any other results related to) unexplained cases.

Main questions:

1. "The 66,571 P/LP variants are found on 8,110,001 alleles in 807,162 individuals, resulting in an average of 10.0 pathogenic alleles per individual" This is interesting, and others in the field have been discussing the rates of carrier status for rare AR disease. Are you saying that on average, a person might be a carrier for 10 different diseases? Anything worth expanding on here? Are more common P/LP variants (CFTR:F508, HFE alleles C282Y and H63D, TTR V142I) included here?

2. In terms of direct application to variant interpretation, if a variant analyst identified a rare variant but it was present in gnomAD, it sounds like you are suggesting that the analyst do all the analyses that you did to determine if there is a reason for this. Using your metrics to calculate if it is likely or unlikely LOF, if the variant is still determined to be likely LOF, and there is no explanation for its presence in gnomAD, what expectation do you have for the AC of the variant? Would you expect it to still be truly absent from gnomAD to require support pathogenicity? I think maybe allele counts for the unexplained variants are in Table S5, but it is hard to tell from the PDF version. If they are not in there, it might be useful to discuss what an "acceptable" AC is likely to be. I know that this is dependent on many other criteria, but it would be helpful to give empirical evidence of how often you see these variants that are unexplained. I do appreciate your mention and conclusion of "overinterpreting pathogenicity in clinic and in research" as I agree that's the concern here and others need to keep in mind that unexplained incomplete penetrance is rare for these disease genes.

3. Can the authors hypothesize or provide additional evidence for anything about specific genes? For example, KMT2C and ARID1B have very low unexplained counts vs. ASXL1, KMT2D and others. SETD2 also looks like >50% unexplained. Do certain genes have a higher % unexplained?

4. Data availability: Variant data is downloadable, but individual level sequencing read data is not, correct? Could you please mention if/when this might cause limitations to interpretation?

Minor points:

Page 4: "The gnomAD dataset has played a key role in supporting the discovery of genes and variants association with genetic diseases". Should this be "associated"?

Page 9: "We used de novo rate as a proxy for penetrance (Table S4)." Is this actually the last column of table S3?

Table S4: formatting issues; some words hidden, some lines missing, "Splice variant were SpliceAI predicts", should this be "where"?

Reviewer #3

(Remarks to the Author)

The manuscript from Gudmundsson et al. is a well written and important paper. There are a number of noteworthy results and it follows on well from previous work. I completely agree with the authors that deep assessment on a variant-by-variant basis is critical, and this paper provides strong evidence to support this assertion. I have only a few minor suggestions.

1. The analysis of missense P/LP variants being modified by pLOF variants in the same gene is particularly novel and interesting, but could benefit from being expanded. There is currently just a single example. Is it possible to estimate what proportion of the 3957 P/LP variants are in cis with the pLOF? Note that the DDG2P database includes information on disease mechanism - are genes with confirmed AD GoF or DM mechanisms over-represented in this group?

2. It is unclear to me how frame-restoring indels were assessed. Did they have to be on the same read as the frameshifting indel? Perhaps the authors could comment on the benefits and challenges of developing a VEP that would simultaneously evaluate the consequence of multiple nearby variants simultaneously, including MNVs and indels.

3. The statistical comparison of gnomAD and ClinVar is very comprehensive, and the finding that every individual carries an average of 10 pathogenic alleles is noteworthy, and could perhaps be highlighted in abstract,

albeit with the caveat that many are in AR genes or have explanations.

4. Please add the proportion of pLOF variants explained by NMD escape rules into the text. This is a well known exclusion from pLOF, and such variants are rarely considered pathogenic in HI conditions.

5. Please add a statistical summary to the text to support the assertion regarding skewed age distribution, rather than requiring readers to dig into the plots in Supp Fig 7.

6. It is hard to tell from a PDF table, but I think variants on X chromosome were excluded. Please comment on this in the text if true.

7. Fig3,C. I note ASXL1 has a particularly high proportion of unexplained pLOFs. There is evidence in the literature of a high level of somatic mosaicism in this gene, which may be worth commenting on or checking allele balance/age of carriers again in this specific case.

8. Fig3,D. This figure suggests that the only NMD escape rule considered was last exon (presumably +55bp) - is that correct? What about variants in the first ~150bp rescued by translation re-initiation? Or variants in >400bp middle exons, where it has also been suggested they may escape NMD.

Version 1:

Reviewer comments:

Reviewer #2

(Remarks to the Author)

Updates to the manuscript are appropriate and aid in further understanding for the reader. I support publication.

Reviewer #3

(Remarks to the Author)

The authors have addressed all my comments and I would strongly support the paper's publication in this journal.

NCOMMS-24-34859-T: Exploring penetrance of clinically relevant variants in over 800,000 humans from the Genome Aggregation Database

Point-by-point response to the reviewers'

Reviewer #1 (Remarks to the Author):

- For the main part of the study, the authors focused on pLoF, but the first part on the presence of a LoF which suppresses the effect of a missense gain of function variant is highly interesting. It would be interesting to have a dedicated part to the missenses MPC>2. For example, if the identified reasons for incomplete penetrance of LoF also explain the predicted deleterious missense variations. I know that the interpretation of the results for the missenses is more complex than for LoF, but as indicated by the authors most of the VUS are missenses and they represent a burden for the clinical geneticists.

The reviewer proposes we expand our analysis described in supplementary Note S1: Rescue by local pLoF events in a subset of P/LP in ClinVar. However, in our analysis of 3957 clinically established deleterious missense variants, only 90 had a pLoF in the same gene of which only one missense variant in *GJB2* gene (highlighted in manuscript) was thought to be an interesting example of rescue by in cis pLoF event, causing incomplete penetrance. As such, there is no evidence that this is a common feature. Given this, we don't believe expanding the analysis to include a larger number of less deleterious missense variants (MPC>2), is likely to change the conclusion in this paper.

This conclusion has already been highlighted in the manuscript:

"Our analysis suggests that this [*GJB2* example] is a rare example of incomplete penetrance as we did not find evidence of this being a common mechanism in the general population (Supplementary report of results in Note S1, Table S3)"

- On page 9, the authors indicate: "We used de novo rate as a proxy for penetrance (Table S4). In these 77 genes, investigating 807,162 individuals, we found 4,464 pLoF variants, of which 3,957 were high-confidence pLoF variants by Loss-Of-Function Transcript Effect Estimator (LOFTEE), 3,223 in exomes (81%), 734 in genomes (19%, Fig. S3)." It is not clear to me if HC-pLoF are more frequent in WES or just that there are more WES data. Sorry for the confusion.

The difference between exomes (81%) and genomes (19%) is mainly due to the volume of data: gnomAD v4 is 9.4% (76,215) genomes and 90.6% (730,947) exomes. Percentage-wise, WGS actually has the higher rate of unexpected pLoF variants.

We have added a sentence to the manuscript to clarify this, page 9, highlighted “gnomAD v4 includes 9.4% (76,215) genomes and 90.6% (730,947) exomes, hence we observed a higher rate of these unexpected pLoF in genomes (19%) compared with exomes (81%) (Fig. S3).”

- The authors have chosen to follow up the 734 HC-pLOF of the WGS. I understand that for some analyses (sQTL) only the WGS can be used, but it would be interesting to estimate the proportion of LoF explained and unexplained for the 3,223 HC-pLOF from the WES.

We were only able to perform the pLoF curation (Figure 3) on a subset of 734 pLoF variants given that it is very labor-intensive analysis, requiring manual variant curation by two independent analysts interpreting each variant based on 39 different criteria. Although not within the scope of this paper, we agree that it would be interesting to investigate if the mode of rescue in the 3,223 HC-pLoF from the WES looks different from GS and if it could add clarity to the overrepresentation of pLoF in genomes vs exomes; 19% of pLoF are found in GS although they only constitute 9% of the gnomAD data. This difference has been highlighted before, for example, it's been shown that genomes have better detection of indels than exomes (Wojcik et al., N Engl J Med, 2024).

- Page 9, “location in a transcript containing a stop-codon” could be replaced by “location in a transcript isoform containing a stop-codon”.

Thank you for this suggestion. We have updated the manuscript accordingly.

- Page 10, the term “evasion” appears (“The observed pLoF evasion rate of 69.8%”). I have an idea of what the authors mean by “evasion” but it would be helpful to have a clear definition.

Thank you for pointing this out. We agree the terminology can be made clear, therefore we have added to the definition in the introduction as follows:

Before

“The framework presented a set of 32 rules designed based on previously accepted pLoF evasion mechanisms and artifact modes to assess the presence of local modifying variants, the biological relevance of the site, and evidence of a variant being an artifact”

Now

“The framework presented a set of 32 rules designed based on previously accepted mechanisms where a variant annotated as pLoF by VEP does not result in loss of the protein product, here referred to as pLoF evasion mechanisms. This includes identifying local modifying variants, assessing the biological relevance of the site, but also evaluating

for evidence of a variant being an artifact”

- The use of the PEXT score is very useful and it would be interesting to have more details on what could have been flagged/filtered out if only brain transcripts were used in the calculation of the PEXT scores. As the authors indicated, there is a lack of prenatal tissue to detect exons specific during fetal life. It would be interesting to test if there are such “prenatal specific exons” in the 77 studied genes and if some of the “explained” pLoF could fall in such exons with low PEXT score in adults and therefore could be miscategorized.

We agree that this is an interesting area to explore further, and these questions are part of a project where we aim to generate embryonic pext scores in collaboration with the Developmental Genotype-Tissue Expression (dGTE_x) Project. However, this analysis will not be included in this manuscript as it’s not within the scope of this article and the data (pext for embryonic tissue) is not available yet.

- The discussion is very well done and address many key aspects related to incomplete penetrance.

Figure 2.

- It might be more precise to add introns and exons in the schematic description of a gene. This is even more important when sQTL are presented in the following figures

Thank you, we agree and have updated Figure 3 and 4 to include introns. For Figure 2, *GJB2* only has one exon and the figure therefore remains the same.

- I would add a label such as “dominant negative mutation” for the Gly45Glu variation.

Thank you, we have updated accordingly, Figure 2a now includes “dominant negative” and “pLoF”.

Figure 3.

- I would suggest to split the panel D in two. One panel with the pLoF with a single explanation and one allowing a better view of the “combination of explanations”.

We agree this is a data-heavy panel. We have added the more granular upset plot version of this result as a supplementary Figure S5 to address this. We have not added a more granular or split figure to Figure 3D as we think that the findings do not justify the space it would require as a main figure.

Second, it is not clear to me if “other tx rescue”, “SpliceAI rescue” or “SpliceAI no Loss” are independent. They all concern 4.7% of the pLoF but they are annotated as a single explanation?

These categories are independent and by chance, they consist of the same number of pLoF. We have clarified this by (1) adding supplementary Figure S5 which shows 24 variants in all three categories and (2) adding the “4.7%” in the text when listing the reasons for escape in the result section.

Reviewer #2 (Remarks to the Author):

Main questions:

1. “The 66,571 P/LP variants are found on 8,110,001 alleles in 807,162 individuals, resulting in an average of 10.0 pathogenic alleles per individual” This is interesting, and others in the field have been discussing the rates of carrier status for rare AR disease. Are you saying that on average, a person might be a carrier for 10 different diseases? Anything worth expanding on here? Are more common P/LP variants (CFTR:F508, HFE alleles C282Y and H63D, TTR V142I) included here?

As described, this analysis included *all* single nucleotide variants and indels (<50 base pairs) reported as P/LP in ClinVar. We have not excluded any variants or genes. Hence, this analysis includes variants associated with adult-onset, mild, or even benign phenotypes (reported as P/LP), for example, carriership for certain blood phenotypes, etc. Hence, the 10.0 pathogenic alleles per individual should *not* be interpreted as carrier status for 10 rare AR disorders per se. Although no further filtering to only include rare disease alleles has been performed, we believe it adds context as to what to expect from a straightforward “all path ClinVar analysis” will show on average. Further, figure 1F right bar adds context for disorders where inheritance pattern has been reported for the gene the variant falls in.

2. In terms of direct application to variant interpretation, if a variant analyst identified a rare variant but it was present in gnomAD, it sounds like you are suggesting that the analyst do all the analyses that you did to determine if there is a reason for this. Using your metrics to calculate if it is likely or unlikely LOF, if the variant is still determined to be likely LOF, and there is no explanation for its presence in gnomAD, what expectation do you have for the AC of the variant? Would you expect it to still be truly absent from gnomAD to require support pathogenicity? I think maybe allele counts for the unexplained variants are in Table S5, but it is hard to tell from the PDF version. If they are not in there, it might be useful to discuss what an “acceptable” AC is likely to be. I know that this is dependent on many other criteria, but it would be helpful to give empirical evidence of how often you see these variants that are unexplained. I do appreciate your mention and conclusion of “overinterpreting pathogenicity in clinic and in research” as I agree that’s the

concern here and others need to keep in mind that unexplained incomplete penetrance is rare for these disease genes.

Presence in gnomAD, at a very low AC, is generally not a reason to discard a variant thought to be pathogenic in variant interpretation analysis. Beyond reduced penetrance, late-onset, mild phenotypes, disease positive individuals in biobanks, etc. there are many additional reasons why a variant can be present in gnomAD (rescues), which is what this manuscript is aiming to address. Although we can show many examples of rescues and explain a high proportion of the unexpected variants, there are likely modes that are not included in our analysis.

Since rescues are very rare, AC is in general very low for severe, early-onset highly penetrant conditions not expected to be found in gnomAD. Although we're not providing a hard cut-off (and do not think a single cut off would be appropriate), we note that the vast majority (almost all) unexpected pLoF will have an AC <5 (Fig. S4) but there are rare instances of higher AC, therefore we refrain from suggesting hard AC cut-offs. Table S5 does not contain AC but Figure S9 shows the number of pLoF remaining in each gene after the full filtering process (Figure 1B, 50 of which 33 are in MANE Select, colored in red in Figure S9a), aka ("unexplained" pLoF variants). As displayed there, most genes investigated have 0 variants if considering all rescue/error modes we describe in the article, and at most a single gene has 6 pLoF variants. And of the 50 variants the majority, n=40, are unique variants all with AC=1.

3. Can the authors hypothesize or provide additional evidence for anything about specific genes? For example, KMT2C and ARID1B have very low unexplained counts vs. ASXL1, KMT2D and others. SETD2 also looks like >50% unexplained. Do certain genes have a higher % unexplained?

There are several gene-specific properties that can result in the inflation of unexpected variants. For example, *ASXL1* has a high number of variants due to somatic variants at high allele balance, as mentioned in the manuscript "*ASXL1 has a pLI of 0.0 in both v4 and v2 due to somatic variants rising to higher allele balance due to clonal hematopoiesis and deflating the pLI constraint score.*" Somatic mosaicism likely accounts for most pLoF in this and other genes affected by clonal hematopoiesis (see supplementary figure 7). Another reason is if a gene has an exon part of an isoform that is not expressed in disease-relevant tissue/timepoint and hence does not cause disease. Like *MEF2C* (Figure 4c) shows an example of this with pLoF clustering in a specific part of the gene that also shows a reduced pext score. Our protocol on pLoF curation further expands on this subject and gives great detail to what rescue modes result in unexpected pLoFs in gnomAD and other databases (DOI: [10.1016/j.ajhg.2023.08.005](https://doi.org/10.1016/j.ajhg.2023.08.005)).

4. Data availability: Variant data is downloadable, but individual level sequencing read data is not, correct? Could you please mention if/when this might cause limitations to interpretation?

Rescue modes described in this article and the LoF curation protocol (DOI: [10.1016/j.ajhg.2023.08.005](https://doi.org/10.1016/j.ajhg.2023.08.005)) do not require individual-level sequencing data, they are found within close proximity of the variant and we have used the read-data that is publicly available on the browser for each variant. The sQTL analysis and the investigation of pLoF as a rescue for gain-of-function variants require individual-level data but are not part of the pLoF interpretation protocol.

Minor points:

Page 4: “The gnomAD dataset has played a key role in supporting the discovery of genes and variants associated with genetic diseases”. Should this be “associated”?

Yes, thank you. We have updated the manuscript accordingly.

Page 9: “We used de novo rate as a proxy for penetrance (Table S4).” Is this actually the last column of table S3?

Yes, thank you. We have updated the references throughout.

Table S4: formatting issues; some words hidden, some lines missing, “Splice variant were SpliceAI predicts”, should this be “where”?

Thank you. We’ll make sure this is correct and readable.

Reviewer #3 (Remarks to the Author):

1. The analysis of missense P/LP variants being modified by pLOF variants in the same gene is particularly novel and interesting, but could benefit from being expanded. There is currently just a single example. Is it possible to estimate what proportion of the 3957 P/LP variants are in cis with the pLOF? Note that the DDG2P database includes information on disease mechanism - are genes with confirmed AD GoF or DM mechanisms over-represented in this group?

Only 90 of the 3957 had a pLoF in that gene (not accounting for phase). Of these 90, only one interesting example (*GJB2* case highlighted in manuscript) remained after assessing according to three criteria: 1) carriership of a pLoF variant in >50% individuals with the P/LP variant (2) a pLoF variant determined as a true LoF resulting in ablated protein product using LoF curation explained below (filtering artifacts, somatic variants, missanotations or rescued variant), (3) a P/LP variant acting through dominant gain-of-function mechanism. It is possible to look at phase (paired read when close, or statistical phasing if present in >5 individuals). We used read data/paired reads for the *GJB2*-example. The result of the analysis has been expanded on in Supplementary Note S1.

2. It is unclear to me how frame-restoring indels were assessed. Did they have to be on the same read as the frameshifting indel?

Yes, they had to be within the same read. We look at the read window provided on the gnomAD browser, around ~200 bp, usually, the indels are just a few base pairs from each other. We refrain from describing the complex details of the pLoF protocol used and instead reference the original article that thoroughly explains the elaborate process of manual pLoF curation: <https://doi.org/10.1016/j.ajhg.2023.08.005>. However, the rules used in this work are also shared in Supplementary Table S4.

Perhaps the authors could comment on the benefits and challenges of developing a VEP that would simultaneously evaluate the consequence of multiple nearby variants simultaneously, including MNVs and indels.

Part of the motivation for this study was to identify the needs for an updated “Loss-Of-Function Transcript Effect Estimator” (LOFTEE) tool. Although this is beyond the scope of this work, there is interest to further develop LOFTEE. High-throughput analysis with the current LOFTEE VEP-plugin provides some information for pLoF variant interpretation but an updated version that accounts for modes presented in this article (including MNVs and indels) will be useful.

3. The statistical comparison of gnomAD and ClinVar is very comprehensive, and the finding that every individual carries an average of 10 pathogenic alleles is noteworthy, and could perhaps be highlighted in abstract, albeit with the caveat that many are in AR genes or have explanations.

The 10 P/LP variants / individual in gnomAD include all P/LP ClinVar variants (including cancer predisposition variants and other subclinical phenotypes linked to variants labeled as P/LP in ClinVar). Hence, the number is not specifically linked to carrier status for AR disorders nor disease risk alleles. Therefore, we refrain from highlighting this number in the abstract and comment on this in the main text where details of the analysis are also given to reduce the risk of misinterpretation.

4. Please add the proportion of pLOF variants explained by NMD escape rules into the text. This is a well-known exclusion from pLOF, and such variants are rarely considered pathogenic in HI conditions.

Yes, thank you. We have added the percentage of all the listed most common rescue modes and added the allele balance of alternate alleles below 25% which had accidentally been left out of listed modes in the result section.

5. Please add a statistical summary to the text to support the assertion regarding skewed age distribution, rather than requiring readers to dig into the plots in Supp Fig 7.

Thank you for this suggestion, we have established the skewed distribution with a Wilcoxon rank sum test and slightly rephrased this section, see below

“For the remaining 106 variants in 195 individuals, we observed a trend of higher age distribution in these 195 individuals compared to gnomAD genomes in general (Fig. S8a-b) consistent with somatic origin which can rise to a high AB due to age-related clonal hematopoiesis. Variants noted to have an appreciable AB but still below the AB for typical germline variants, defined as an alternative allele ratio under 35% (with those under 20% already filtered by gnomAD QC practices and below 25% filtered by pLoF curation protocol) showed a significant skew towards elderly individuals (two-sided Wilcoxon rank sum test, $p = 0.0013$, Fig. S8c), which was also observed for variants in clonal hematopoiesis-associated genes (36 genes²⁹; two-sided Wilcoxon rank sum test, $p = 0.00027$, Fig. S8d)”

6. It is hard to tell from a PDF table, but I think variants on X chromosome were excluded. Please comment on this in the text if true.

Yes, for 77 genes linked to severe disorders, we filter to only include genes associated with disorders of autosomal dominant inheritance pattern (as stated in results and more in details in methods), hence X and Y chromosomes are excluded. The supplementary tables will be provided as an Excel file in the published version for improved readability.

7. Fig3,C. I note ASXL1 has a particularly high proportion of unexplained pLOFs. There is evidence in the literature of a high level of somatic mosaicism in this gene, which may be worth commenting on or checking allele balance/age of carriers again in this specific case.

Yes, *ASXL1* has a high number of variants due to somatic variants at high allele balance, as mentioned in the manuscript “*ASXL1 has a pLI of 0.0 in both v4 and v2 due to somatic variants rising to higher allele balance due to clonal hematopoiesis and deflating the pLI constraint score.*” Somatic mosaicism likely accounts for most pLoF in this and other genes affected by clonal hematopoiesis (as shown in Supplementary Figure 7).

8. Fig3,D. This figure suggests that the only NMD escape rule considered was last exon (presumably +55bp) - is that correct?

“Last exon” category also includes the last 50 base pairs of the penultimate exon as stated in the manuscript “the last exon (or last 50 base pairs of the penultimate exon)” and as defined by Singer-Berk et al.. <https://doi.org/10.1016/j.ajhg.2023.08.005>. Also, details of the rules used in this work are shared in Supplementary Table S4, cell E6.

What about variants in the first ~150 bp rescued by translation re-initiation?

Variants in the first ~150 bp that have an alternate Methionine downstream of the stop variant (within the first exon) that could rescue are included as rescues in the “other tx rescue”. We refrain from describing details of the protocol and instead referenced Singer-Berk et al., <https://doi.org/10.1016/j.ajhg.2023.08.005>. However, we understand that this might be an important detail and we have therefore added to the manuscript to clarify

“other transcript rescues (including transcripts with downstream methionine within the first exon, unconserved alternate open reading frame, and variants in overhang exons 4.7%)²⁵,”

Variants in first 150bp *without* a downstream methionine are not labeled as “not LoF”/rescued but have been noted throughout the curation process. Specifics on the number of LoF/likely LoF variants in this category can be found in Supplementary Figure S8.

Or variants in >400bp middle exons, where it has also been suggested they may escape NMD.

As for 150bp *without* a downstream methionine, variants in exons larger than 400 bp were monitored during the curation process but as of now have not been labeled as not pLoF/rescued, details in Supplementary Figure S8. We do note that 8 of 33 unexplained pLoF in MANE Select transcripts occurred in long exons which is interesting and likely an overrepresentation (Supplementary Figure S8C).

Additional updates

In addition to updates requested by the reviewers we have

- Added a “table of contents” to supplementary information to aid navigation through this now large document
- Total counts have been added to Supplementary Table S6 to ease navigation of the table
- Formatting issues with data points in table S8 have been corrected (probably occurred due to converting to excel file)
- Updated to text (page 10, highlighted) regarding Figure 3G: 122/734 variants were reported in ClinVar but only variants with B/LB/VUS/LP/P classification were included in analysis, that is 108/122 variants. 122 have been updated to 108 which is what was reported in figure 3G. Remaining 14 variants were of conflicting interpretation, they are included in Supplementary Fig. S7.
- The competing interest statement for H.L.R was updated
- For improved flow a section on page 6 starting with “Although the vast majority of P/LP variants in” have been moved to the previous paragraph. The text is still identical.
- We updated typos and grammatical errors.